# Implicit Denoiser Structure in Robust Classifiers Explains Generative Capabilities

## Abstract

Adversarially robust neural networks, while designed for classification, exhibit surprising generative capabilities when appropriately probed. We provide a theoretical framework explaining this phenomenon by connecting adversarial robustness to implicit denoising structure. Building on established results that robust training drives Jacobians toward low-rank solutions, we demonstrate that the Gram operator $\mathbf{J}^\top \mathbf{J}$ functions as an implicit denoiser, selectively preserving signal along discriminative subspaces while suppressing noise in orthogonal directions. This insight leads to Prior-Guided Drift Diffusion (PGDD), a simple algorithm that leverages this structure for generation through inference objectives rather than explicit Jacobian computation. PGDD requires no generative training or architectural modifications, yet produces class-consistent samples across different datasets and architectures. We extend our approach to standard networks via sPGDD, demonstrating that implicit generative structure exists beyond adversarially trained models. Our results establish a connection between discriminative robustness and generative modeling, showing that robust classifiers encode statistical priors that enable structured pattern generation without explicit generative objectives.

## 1 Introduction

Adversarial training has emerged as a critical defense mechanism for ensuring the safety and reliability of neural networks deployed in high-stakes applications, where robustness to input perturbations is essential for preventing adversarial attacks and maintaining system integrity (Madry et al., 2018; Wong et al., 2020). Originally developed to address security vulnerabilities in machine learning systems (Goodfellow et al., 2015; Carlini & Wagner, 2017), adversarial training has revealed unexpected emergent properties: robust models can function as implicit generative models and produce structured images when appropriately probed (Santurkar et al., 2019; Engstrom et al., 2019). This dual discriminative–generative behavior suggests that the mechanisms underlying adversarial robustness may be more fundamental than previously recognized, yet the theoretical foundations of these emergent capabilities remain largely unexplored.

Recent theoretical advances have begun to illuminate the mathematical structure underlying adversarial robustness. Studies have demonstrated that the spectral properties of neural networks such as input-output Jacobians are directly linked to generalization and robustness (Oymak et al., 2019; Wu & Li, 2024). This spectral properties force networks to suppress sensitivity along most input directions while preserving discriminative power along a small subspace (Hoffman et al., 2019; Jakubovitz & Giryes, 2018). Jacobian regularization techniques have formalized this connection, showing that controlling gradient norms directly improves robustness by constraining the Jacobian spectrum (Ross & Doshi-Velez, 2017; Rodríguez-Muñoz et al., 2025). However, despite these insights into the *discriminative* implications of spectral structure, the potential *generative* consequences of low-rank Jacobians remain largely unexplored.

In this work, we bridge this gap by drawing inspiration from the success of denoising diffusion probabilistic models (DDPMs), which achieve remarkable generative performance through learned denoising operations (Ho et al., 2020; Song & Ermon, 2019). The main related insight from diffusion models is that networks trained to remove noise implicitly learn the score function of the data distribution, enabling iterative generation through gradient-based sampling (Song et al., 2021). This connection between denoising and generation motivates our central hypothesis: the

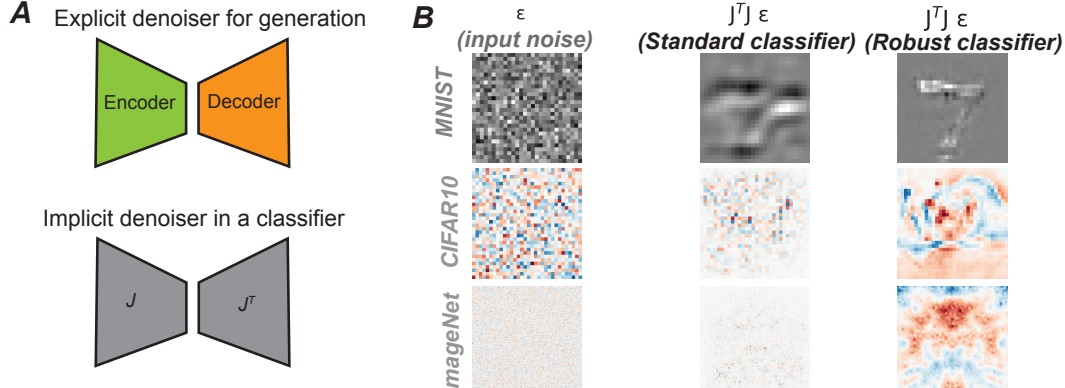

Figure 1: **Adversarial robustness creates implicit denoisers through $\mathbf{J}^\top\mathbf{J}$ structure.** (A) Explicit denoisers use separate encoder-decoder architectures for noise removal and generation (Vincent et al., 2008; Vincent, 2011) . Our hypothesis: robust classifiers develop mathematically equivalent structure where the Jacobian $\mathbf{J}$ and its transpose $\mathbf{J}^\top$ naturally form an implicit denoising operator $\mathbf{J}^\top\mathbf{J}$. (B) Empirical validation on MNIST, CIFAR10, and ImageNet: when $\mathbf{J}^\top\mathbf{J}$ is applied to input noise $\epsilon$, standard classifiers fail to reject the random structure, while robust classifiers extract shape-like patterns, demonstrating denoising capability. Images are individually normalized to reveal structure (more details in Supp. C.1).

low-rank structure induced by adversarial training makes the Gram operator $\mathbf{J}^\top\mathbf{J}$ function as an implicit denoiser, selectively preserving discriminative directions while suppressing noise along orthogonal subspaces. Just as DDPMs leverage explicit denoising networks for generation, we demonstrate that robust classifiers contain implicit denoising structure that can be exploited for the same purpose. We introduce *Prior-Guided Drift Diffusion (PGDD)*, an algorithm that harnesses this hidden structure through inference objectives rather than explicit Jacobian computation, enabling practical generation from robust classifiers. We further develop *sPGDD*, a variant that extends our approach to standard networks through gradient smoothing techniques. It is important to note, in contrast to earlier approaches—which require a target class label or reference objective and therefore perform conditional synthesis—PGDD performs *unconditional* generation that arises directly from the spectral properties of the network.

This work makes several key contributions to understanding the connection between adversarial robustness and generative modeling:

- This work establishs that the Gram operator $\mathbf{J}^\top\mathbf{J}$ in adversarially robust classifiers functions as an implicit denoiser, connecting prior findings on low-rank Jacobian structure to generative capabilities. This provides the first principled explanation for why robust classifiers exhibit emergent generative properties.

- It also demonstrate through spectral analysis, energy ratio measurements, and visual residuals that robust classifiers suppress noise while amplifying class-consistent structure.

- We introduce Prior-Guided Drift Diffusion (PGDD), a practical algorithm that leverages implicit $\mathbf{J}^\top\mathbf{J}$ structure for generation through inference objectives rather than explicit Jacobian computation. PGDD requires no architectural modifications or generative training.

- Finally, we develop sPGDD (smooth PGDD), which enables generative inference in standard classifiers through gradient smoothing techniques, demonstrating that implicit generative structure exists beyond robust networks.

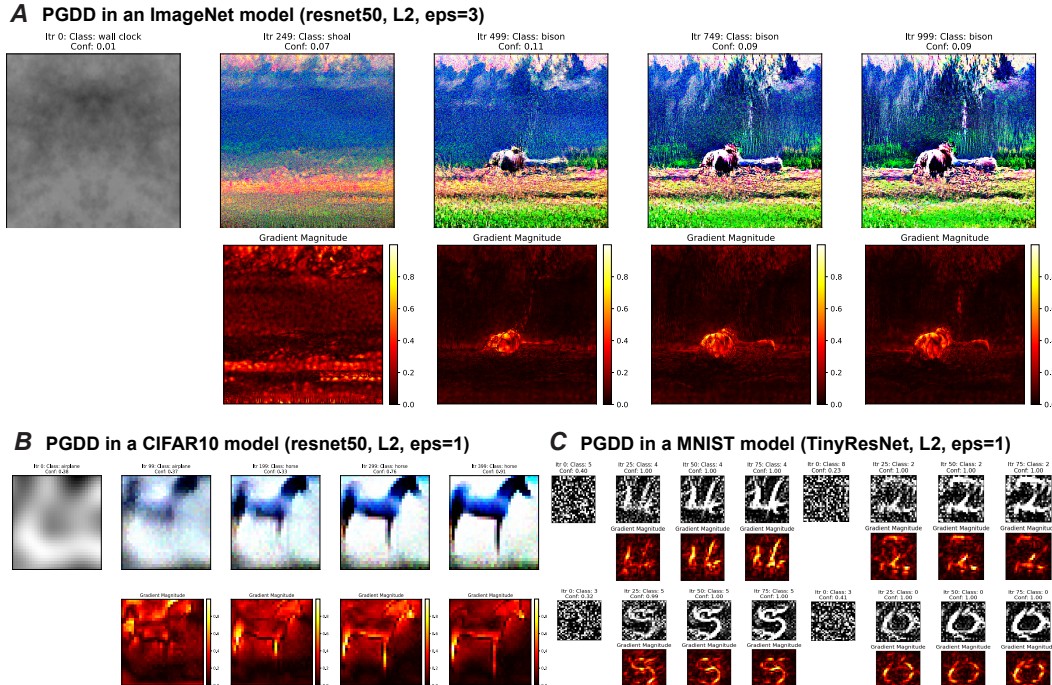

Figure 2: **Prior-Guided Drift Diffusion (PGDD) generates coherent images from input noise using robust classifiers.** PGDD applied to **A** a robust ResNet-50 (ImageNet, $\ell_2$ adversarially trained with $\epsilon = 3, 5$) **B** a robust ResNet-50 (CIFAR10, $\ell_2$ adversarially trained with $\epsilon = 1, 5$) **C** a robust TinyResNet (MNIST, $\ell_2$ adversarially trained with $\epsilon = 1, 5$) demonstrates progressive refinement from random noise (same for both shown trajectories) to semantically coherent images. Starting from noise (Itr 0), the algorithm iteratively moves away from a noisy representation of the original input (see algorithm in 1). No explicit generative training was used, only PGDD on pretrained adversarially robust classifiers (Supplementary F.1).

## 2 IMPLICIT DENOISER IN ROBUST CLASSIFIERS

### 2.1 GENERATIVE POWER OF ADVERSARIALLY ROBUST CLASSIFIERS

The observation that adversarially robust classifiers exhibit unexpected generative capabilities has garnered increasing attention across multiple domains. Recent empirical work has demonstrated that robust models can synthesize structured images (Santurkar et al., 2019), produce perceptually aligned gradients Kaur et al. (2019), and exhibit improved correspondence between their internal representations and human-perceivable features (Engstrom et al., 2019). Intriguingly, these generative properties emerge without explicit generative training objectives, suggesting an intrinsic connection between discriminative robustness and generative modeling capacity.

Previous work has explored connections between classifiers and generative models, but these approaches typically require training classifiers with explicit generative objectives. Joint Energy-based Models (JEMs) train networks to simultaneously perform classification and generation by optimizing both discriminative and generative losses (Grathwohl et al., 2020). Similarly, gradient alignment methods improve model interpretability by training the implicit density model to align with ground truth distributions (Singla et al., 2021). However, these methods fundamentally alter the training process to achieve generative capabilities, whereas robust classifiers exhibit these properties as emergent byproducts of adversarial training alone.

Despite these compelling empirical demonstrations, the theoretical foundations underlying the generative capacity of robust classifiers remain largely unexplored. While extensive work has characterized the links between spectral properties and robustness (Hoffman et al., 2019; Jakubovitz & Giryes, 2018; Oymak et al., 2019; Wu & Li, 2024), no principled framework has emerged to explain

how these spectral characteristics translate to generative functionality. Several lines of research have provided crucial building blocks for our theoretical development. Studies on Jacobian regularization have established that controlling gradient norms enhances robustness to adversarial perturbations (Hoffman et al., 2019; Jakubovitz & Giryes, 2018). Complementary work has demonstrated that adversarial training fundamentally alters the spectral properties of neural networks (Du et al., 2019; Sinha et al., 2018). However, these findings have been studied in isolation, without connecting spectral structure to generative inference capabilities.

The success of denoising diffusion probabilistic models (DDPMs) provides additional theoretical context. DDPMs achieve remarkable generative performance by learning to reverse a noise process, with the core insight that score functions can guide iterative denoising Ho et al. (2020); Song & Ermon (2019). This raises a natural question: could robust classifiers similarly encode implicit score functions that enable generative inference?

## 2.2 THEORETICAL FRAMEWORK: SIGNAL-NOISE DECOMPOSITION IN $\mathbf{J}^\top \mathbf{J}$

**Theorem 2.1** (Signal–Noise Decomposition in $\mathbf{J}^\top \mathbf{J}$). *Let $f : \mathbb{R}^P \to \mathbb{R}^K$ be a neural classifier with Jacobian $\mathbf{J}(\mathbf{x})$ at input $\mathbf{x}$, and let the Gram operator admit the eigendecomposition $\mathbf{J}^\top \mathbf{J} = \mathbf{V}\mathbf{\Lambda}\mathbf{V}^\top$ with eigenvalues $\mathbf{\Lambda} = \mathrm{diag}(\lambda_1, \ldots, \lambda_P)$ and orthonormal eigenvectors $\mathbf{V} = [\mathbf{v}_1, \ldots, \mathbf{v}_P]$. For any perturbation $\boldsymbol{\epsilon} = \sum_{i=1}^P c_i \mathbf{v}_i$, we have*

$$\mathbf{J}^\top \mathbf{J} \boldsymbol{\epsilon} = \sum_{i=1}^P \lambda_i c_i \mathbf{v}_i, \tag{1}$$

$$\|\mathbf{J}^\top \mathbf{J} \boldsymbol{\epsilon}\|_2^2 = \sum_{i=1}^P \lambda_i^2 c_i^2. \tag{2}$$

*Suppose adversarial training induces spectral concentration such that there is a small set of "signal" directions $\mathcal{S} = \{1, \ldots, k\}$ with $k \ll P$ and*

$$\lambda_1 \gg \lambda_2 \gg \cdots \gg \lambda_k \gg \lambda_{k+1} \approx \cdots \approx \lambda_P \approx 0.$$

*Define the decomposition $\boldsymbol{\epsilon} = \boldsymbol{\epsilon}_\parallel + \boldsymbol{\epsilon}_\perp$ where $\boldsymbol{\epsilon}_\parallel = \sum_{i \in \mathcal{S}} c_i \mathbf{v}_i$ and $\boldsymbol{\epsilon}_\perp = \sum_{i \notin \mathcal{S}} c_i \mathbf{v}_i$. Then for isotropic noise $\boldsymbol{\epsilon} \sim \mathcal{N}(0, \sigma^2 \mathbf{I}_P)$,*

$$\mathbb{E}\Big[\|\mathbf{J}^\top \mathbf{J} \boldsymbol{\epsilon}_\parallel\|_2^2\Big] = \sigma^2 \sum_{i \in \mathcal{S}} \lambda_i^2, \qquad \mathbb{E}\Big[\|\mathbf{J}^\top \mathbf{J} \boldsymbol{\epsilon}_\perp\|_2^2\Big] = \sigma^2 \sum_{i \notin \mathcal{S}} \lambda_i^2 \approx 0. \tag{3}$$

*In particular, under spectral concentration $\sum_{i \notin \mathcal{S}} \lambda_i^2 \ll \sum_{i \in \mathcal{S}} \lambda_i^2$, we obtain*

$$\mathbb{E}\Big[\|\mathbf{J}^\top \mathbf{J} \boldsymbol{\epsilon}_\parallel\|_2^2\Big] \gg \mathbb{E}\Big[\|\mathbf{J}^\top \mathbf{J} \boldsymbol{\epsilon}_\perp\|_2^2\Big], \tag{4}$$

*i.e., $\mathbf{J}^\top \mathbf{J}$ strongly amplifies or preserves signal-aligned components while suppressing noise components orthogonal to $\mathcal{S}$. Thus the low-rank spectral structure induced by adversarial training naturally endows $\mathbf{J}^\top \mathbf{J}$ with an implicit denoising effect.*

*Proof.* By expanding $\boldsymbol{\epsilon}$ in the eigenbasis $\{\mathbf{v}_i\}$, we have $\boldsymbol{\epsilon} = \sum_{i=1}^P c_i \mathbf{v}_i$ and, since $\mathbf{J}^\top \mathbf{J} \mathbf{v}_i = \lambda_i \mathbf{v}_i$,

$$\mathbf{J}^\top \mathbf{J} \boldsymbol{\epsilon} = \sum_{i=1}^P \lambda_i c_i \mathbf{v}_i, \quad \|\mathbf{J}^\top \mathbf{J} \boldsymbol{\epsilon}\|_2^2 = \sum_{i=1}^P \lambda_i^2 c_i^2.$$

For $\boldsymbol{\epsilon} \sim \mathcal{N}(0, \sigma^2 \mathbf{I}_P)$, the coefficients $c_i$ are independent with $\mathbb{E}[c_i^2] = \sigma^2$. Therefore,

$$\mathbb{E}\Big[\|\mathbf{J}^\top \mathbf{J} \boldsymbol{\epsilon}_\parallel\|_2^2\Big] = \mathbb{E}\Big[\sum_{i \in \mathcal{S}} \lambda_i^2 c_i^2\Big] = \sigma^2 \sum_{i \in \mathcal{S}} \lambda_i^2,$$

and similarly

$$\mathbb{E}\Big[\|\mathbf{J}^\top \mathbf{J} \boldsymbol{\epsilon}_\perp\|_2^2\Big] = \sigma^2 \sum_{i \notin \mathcal{S}} \lambda_i^2.$$

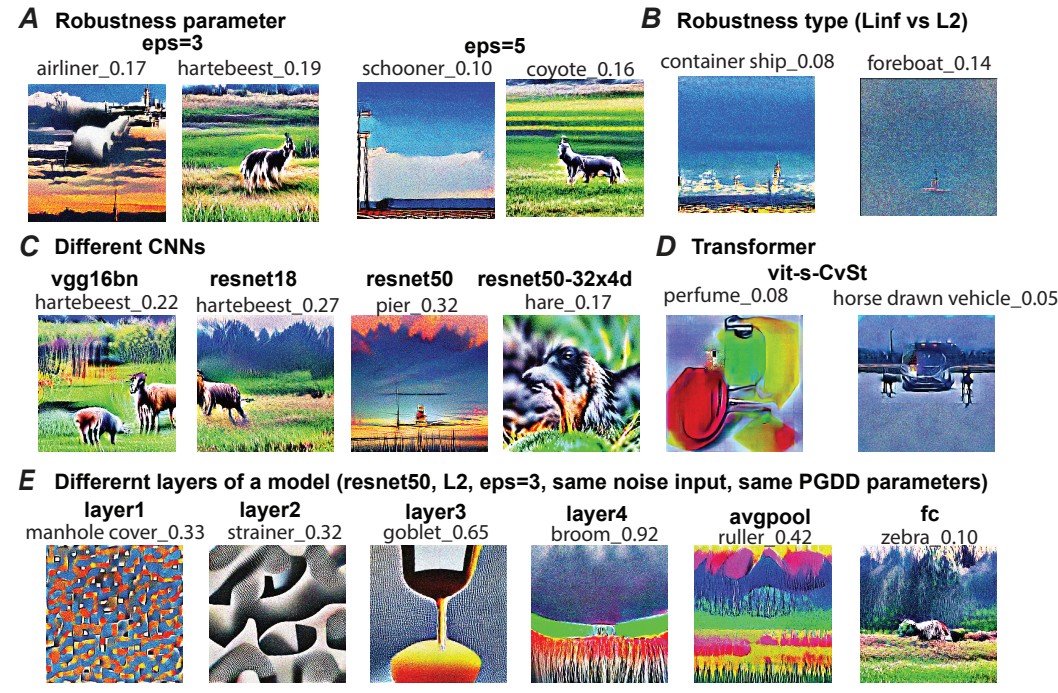

Figure 3: **PGDD across architectures and applied at different internal layers.** PGDD applied to robust ImageNet classifiers demonstrates consistent semantic generation across: (A) Different robustness parameters ($\varepsilon = 5$) showing class-consistent outputs for various ImageNet categories; (B) Different adversarial training norms ($L_\infty$ vs $L_2$) with comparable generation quality; (C) Different CNN architectures (VGG16-BN, ResNet-18, ResNet-50, ResNeXt-50-32x4d) all producing coherent semantic content; (D) Vision Transformer architecture (ViT-s-CvSt (Singh et al., 2023)) demonstrating that implicit denoising structure generalizes beyond CNNs; (E) Different internal layers of ResNet-50 ($L_2$, $\varepsilon = 3$) showing that PGDD can be applied at various representational levels to generate semantically meaningful patterns. All examples use identical noise initialization and PGDD parameters (Supplementary F.1).

Under spectral concentration, the latter sum is negligible compared to the former, yielding

$$\mathbb{E}\left[\|\mathbf{J}^\top \mathbf{J}\boldsymbol{\epsilon}_\parallel\|_2^2\right] \gg \mathbb{E}\left[\|\mathbf{J}^\top \mathbf{J}\boldsymbol{\epsilon}_\perp\|_2^2\right].$$

This shows that $\mathbf{J}^\top \mathbf{J}$ concentrates energy in the discriminative subspace $\mathcal{S}$ and suppresses orthogonal components, providing the claimed signal–noise decomposition and implicit denoising behavior. □

This formalizes our central hypothesis: $\mathbf{J}^\top \mathbf{J}$ suppresses random noise components while preserving structured components aligned with the discriminative subspace $\mathcal{S}$. Crucially, this denoising capability requires no additional training beyond the original robustness objective. Just as explicit denoising autoencoders use encoder-decoder architectures for generation (Figure 1A), robust classifiers naturally develop mathematically equivalent structure where $\mathbf{J}$ and $\mathbf{J}^\top$ form an implicit denoising operator $\mathbf{J}^\top \mathbf{J}$ (Figure 1B). This framework makes several testable predictions that we validate empirically: (1) the energy ratio $\|\mathbf{J}^\top \mathbf{J}\boldsymbol{\epsilon}\|/\|\boldsymbol{\epsilon}\|$ should be much smaller for robust than standard models, (2) applying $\mathbf{J}^\top \mathbf{J}$ to random noise should reveal class-consistent structure in robust classifiers, and (3) robust models should exhibit stronger spectral concentration with higher $\lambda_1/\lambda_2$ ratios and steeper eigenvalue decay.

## 3 Prior–Guided Drift Diffusion (PGDD)

Having established that robust classifiers contain implicit denoising structure through $\mathbf{J}^\top \mathbf{J}$, we now address how to leverage this capability for generation. Previous methods for generating images

using adversarially robust classifiers have employed procedures essentially equivalent to targeted adversarial attacks (Santurkar et al., 2019). These approaches optimize inputs to maximize specific class predictions, effectively using the classifier's gradients to guide generation. We wish to apply the denoising operator $\mathbf{J}^\top \mathbf{J}$ without forming it explicitly.

Our approach is inspired by adversarial purification methods: we first corrupt the input with noise, then attempt to move away from that noisy representation using gradients. Intuitively, this process should reveal the underlying structure that the network has learned to distinguish from corruption. We show that $\mathbf{J}^\top \mathbf{J}$ emerges naturally from this simple inference objective.

For $\epsilon \sim \mathcal{N}(0, \sigma^2 \mathbf{I})$ and layer $r(\cdot)$,

$$\mathcal{L}_{\text{PGDD}}(\mathbf{x}; \epsilon) = \left\| r(\mathbf{x}) - \text{sg}[r(\mathbf{x} + \epsilon)] \right\|_2^2, \tag{5}$$

where $\text{sg}[\cdot]$ stops gradients through its argument.

Let $\mathbf{J}_r(\mathbf{x}) = \nabla_{\mathbf{x}} r(\mathbf{x})$. Then

$$\nabla_{\mathbf{x}} \mathcal{L}_{\text{PGDD}}(\mathbf{x}; \epsilon) = 2 \mathbf{J}_r(\mathbf{x})^\top \big( r(\mathbf{x}) - \text{sg}[r(\mathbf{x} + \epsilon)] \big) \tag{6}$$

Using the first-order Taylor approximation $r(\mathbf{x} + \epsilon) \approx r(\mathbf{x}) + \mathbf{J}_r(\mathbf{x})\epsilon$, we have:

$$\nabla_{\mathbf{x}} \mathcal{L}_{\text{PGDD}}(\mathbf{x}; \epsilon) \approx 2 \mathbf{J}_r(\mathbf{x})^\top \Big( r(\mathbf{x}) - \big( r(\mathbf{x}) + \mathbf{J}_r(\mathbf{x})\epsilon \big) \Big)$$
$$= -2 \mathbf{J}_r(\mathbf{x})^\top \mathbf{J}_r(\mathbf{x}) \, \epsilon. \tag{7}$$

Since the goal is to move away from the noisy representation, we ascend on equation 5. The PGDD algorithm proceeds as follows: we first sample noise $\epsilon \sim \mathcal{N}(0, \sigma^2 \mathbf{I})$, then iteratively update:

$$\mathbf{x} \leftarrow \mathbf{x} + \eta \, \nabla_{\mathbf{x}} \mathcal{L}_{\text{PGDD}}(\mathbf{x}; \epsilon) \approx \mathbf{x} - 2\eta \, \mathbf{J}_r(\mathbf{x})^\top \mathbf{J}_r(\mathbf{x}) \, \epsilon, \tag{8}$$

which applies the denoising step $-\mathbf{J}_r^\top \mathbf{J}_r \, \epsilon$. This update is repeated for $T$ iterations with the same $\epsilon$, progressively moving away from the initial noisy representation. Thus, $\mathbf{J}^\top \mathbf{J}$ emerges naturally from the simple objective of moving away from noisy representations.

### 3.1 SMOOTHED PGDD (sPGDD) FOR STANDARD NEURAL NETWORKS

While prior work emphasizes that only adversarially robust classifiers exhibit strong generative behavior, our theoretical framework predicts that standard networks should nevertheless possess weak but non-zero implicit generative structure. This follows from recent results on implicit gradient regularization in standard classifiers (e.g., (Barrett & Dherin, 2020)), which show that gradient descent induces smoothing and structure in $\nabla_{\mathbf{x}} f(\mathbf{x})$ even without explicit robustness constraints. Combined with our spectral analysis of $J^\top J$, these observations suggest that standard models should contain a faint version of the signal–noise decomposition responsible for denoising in robust networks. However, this structure is too weak to be revealed by PGDD directly, whose single-sample gradients are dominated by noise-sensitive components.

To expose this small but stable structure, we introduce smoothed PGDD (sPGDD), a noise-averaged version of PGDD inspired by SmoothGrad (Smilkov et al., 2017). Whereas SmoothGrad is typically used for saliency visualization, we repurpose noise averaging for generative inference. At each iteration, instead of using a single noisy gradient, we compute

$$g = \frac{1}{n} \sum_{i=1}^{n} \nabla_{\mathbf{x}} \mathcal{L}_{\text{PGDD}}(\mathbf{x}; \epsilon_i), \quad \epsilon_i \sim \mathcal{N}(0, \sigma_{\text{smooth}}^2 \mathbf{I}), \tag{9}$$

with $n$ typically in the range 50–200. Averaging suppresses high-variance directions that fluctuate across perturbations and amplifies the low-dimensional components that are consistently present—those aligned with the weak denoising structure predicted by our theory. The update step mirrors PGDD:

$$\mathbf{x} \leftarrow \mathbf{x} + \eta \frac{g}{\|g\| + 10^{-10}}, \tag{10}$$

optionally combined with a small diffusion term for stability (details in Appendix E.1).

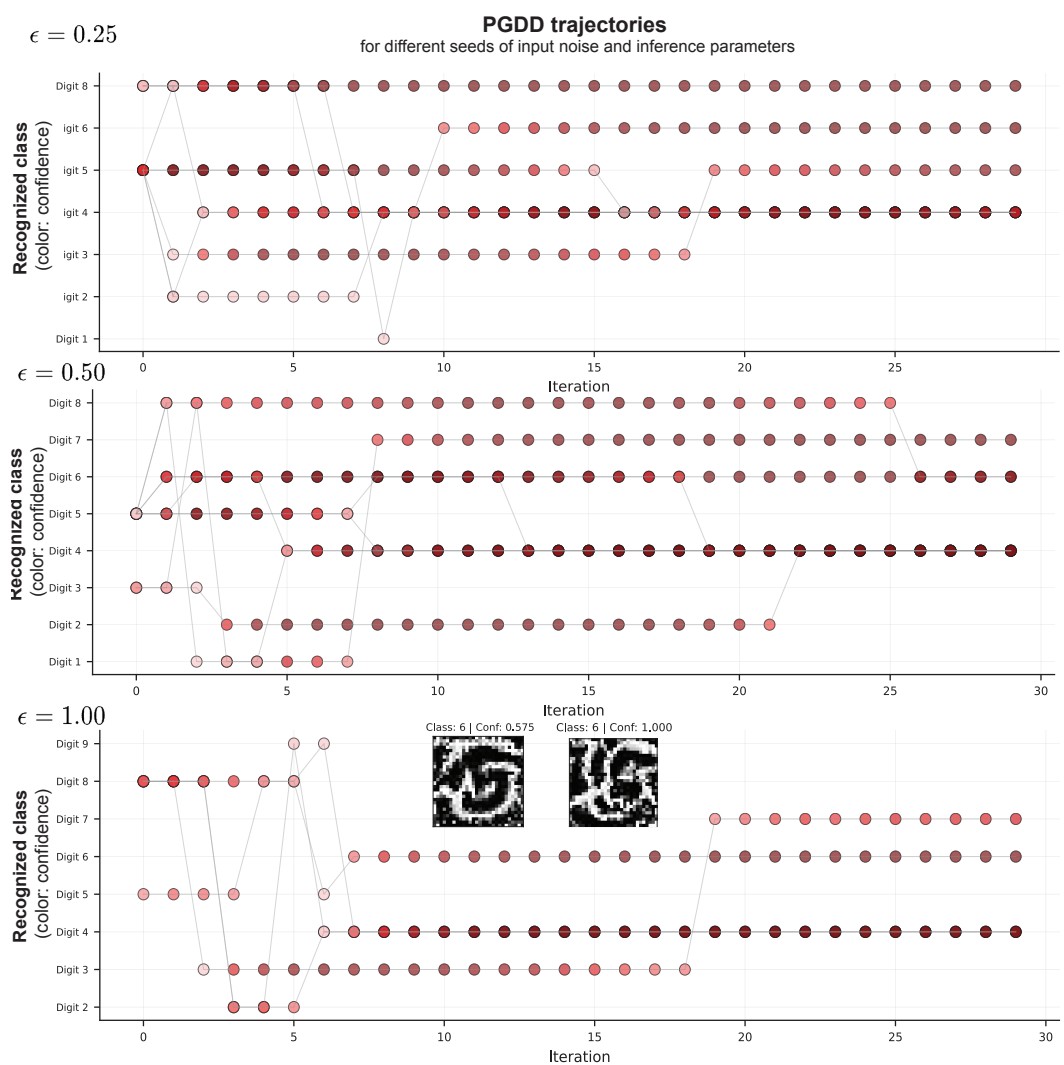

Figure 4: **PGDD exhibits convergent class prediction trajectories across different initializations.** Classification trajectories for PGDD applied to a robust MNIST classifier, starting from different random noise seeds and inference parameters ($L_2$ robust models with top to bottom: $\epsilon = 0.25, 0.5, 1$). Each line represents the predicted class and confidence evolution (depicted in color) during generation. (examples shown: final outputs for class 6 with confidence 1.0 and 0.57).

Empirically, sPGDD yields substantially smoother and more interpretable trajectories in standard networks than PGDD, though with lower fidelity than robust models due to the weaker spectrum of $J^\top J$. Importantly, sPGDD reveals class-consistent structure in standard classifiers, a generative capability not previously reported and does so at a magnitude consistent with our theoretical prediction that implicit regularization produces only a mild denoising effect in non-robust models.

## 4 RESULTS

We evaluate our proposed theoretical framework through two complementary experimental approaches. First, we validate the spectral properties predicted by our theory using MNIST classifiers, where computational tractability allows detailed Jacobian analysis. Second, we demonstrate the practical generative capabilities of PGDD across datasets and architectures, showing that implicit denoising structure enables coherent image generation from robust classifiers. Unless otherwise

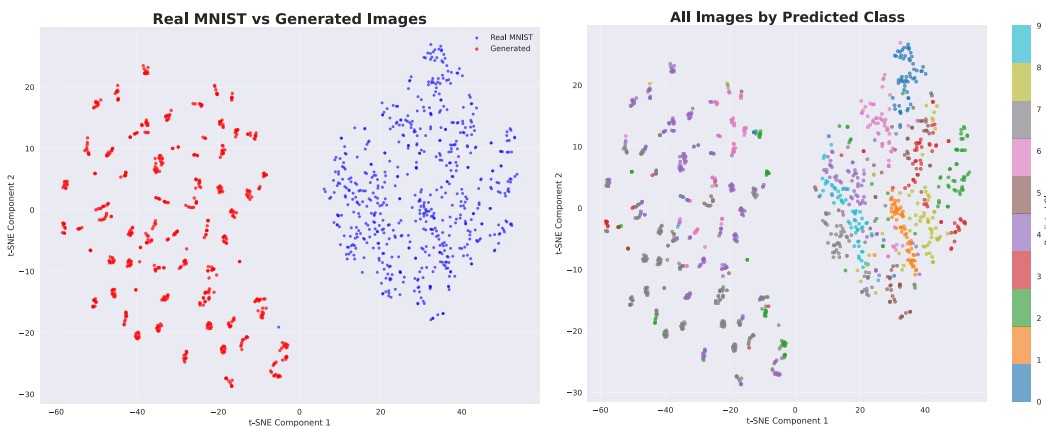

Figure 5: **PGDD-generated images occupy distinct regions in representation space from real MNIST data.** t-SNE visualization of penultimate layer representations comparing real MNIST training images (red) with PGDD-generated samples (blue) from robust classifiers across different initialization seeds for input noise and inference seeds (same PGDD parameters).

stated, $r(\cdot)$ in equation (5) refers to the logits (pre-softmax) layer of the classifier throughout all experiments.

## 4.1 SPECTRAL ANALYSIS: VALIDATING THE IMPLICIT DENOISER HYPOTHESIS

**Notation.** Let $\mathbf{J}(\mathbf{x})$ denote the Jacobian of the classifier at input $\mathbf{x}$, and $\mathbf{G}(\mathbf{x}) = \mathbf{J}(\mathbf{x})^\top \mathbf{J}(\mathbf{x})$ the Gram operator. Let $\lambda_1 \geq \lambda_2 \geq \cdots$ denote its eigenvalues. We define the *energy ratio* as $\|\mathbf{G}(\mathbf{x})\epsilon\|_2/\|\epsilon\|_2$ for $\epsilon \sim \mathcal{N}(0, I)$, and *tail@k* as $\lambda_k$, the $k$-th largest eigenvalue. $R^2$ denotes the coefficient of determination for the quadratic fit $\Delta_{\text{robust}}(\sigma) \approx \alpha\,\sigma^2$.

**Eigenvalue decay and energy ratios.** Eigenvalue analysis confirms that robust models exhibit rapidly decaying Jacobian spectra, with tail eigenvalue suppression showing a $49\times$ reduction (tail@$k$: $8.251 \rightarrow 0.167$). While the leading eigenvalue ratios show modest improvement ($\lambda_1/\lambda_2$: $1.47 \rightarrow 1.96$), the critical denoising mechanism operates through suppression of orthogonal noise directions, as evidenced by the dramatic tail eigenvalue reduction. Our quantitative analysis (Table 1) shows robust classifiers achieve dramatically stronger denoising: energy ratio statistics ($\|\mathbf{J}^\top \mathbf{J}\epsilon\|/\|\epsilon\|$) decrease from 6.19 in standard models to 0.16 in robust models, a $40\times$ improvement. The robust classifier also exhibits a near-perfect fit to the theoretical robustness relationship ($R^2 = 0.992$ vs $0.717$), confirming our framework's predictive accuracy.

Table 1: **Quantitative verification of the implicit denoiser hypothesis.** Robust models show (1) dramatically smaller Jacobian spectrum (reduced $\lambda_1, \lambda_2$, and tail@8), (2) lower energy amplification $\|\mathbf{G}\epsilon\|/\|\epsilon\|$, and (3) stronger quadratic noise response ($R^2$).

| Model | $\lambda_1$ | $\lambda_2$ | $\lambda_1/\lambda_2$ | tail@k | $\|\mathbf{J}^\top \mathbf{J}\epsilon\|/\|\epsilon\|$ | $R^2$ |
|---|---|---|---|---|---|---|
| Standard | 116.8 | 79.57 | 1.47 | 8.251 | 6.19 | 0.717 |
| Robust L2 | 3.166 | 1.612 | 1.96 | 0.167 | 0.16 | 0.992 |
| Ratio | $0.027\times$ | $0.020\times$ | $1.33\times$ | $0.020\times$ | $0.026\times$ | $1.38\times$ |

**Visual residuals reveal class priors.** Applying $\mathbf{J}^\top \mathbf{J}$ to random noise produces dramatically different outcomes in robust versus standard classifiers (Figure 1C). Standard models preserve the random structure of input noise, while robust models extract digit-like patterns aligned with learned class priors. When images are individually normalized to reveal structure, robust models consistently produce recognizable features that correspond to the network's predictions, demonstrating that $\mathbf{J}^\top \mathbf{J}$ functions as both a denoiser and an amplifier of implicit statistical knowledge.

### 4.2 GENERATION WITH PGDD: INSIGHTS INTO CLASSIFIERS' PRIORS

**PGDD generates coherent semantic content from robust models.** Starting from identical noise patterns and using identical inference parameters, PGDD applied to ResNet-50 models (He et al., 2016) trained on ImageNet (Deng et al., 2009) with different $\ell_2$ perturbation budgets ($\epsilon = 3$ vs $\epsilon = 5$) using PGD adversarial training (Madry et al., 2018) converges to semantically distinct but equally coherent patterns—parachute/landscape scenes versus container ship/seascape scenes (Figure 2). These serve as representative instances of PGDD trajectory convergence on ImageNet-trained priors, demonstrating that different robustness constraints create distinct semantic attractors. We further validated our approach by testing sPGDD on standard ResNet-50 and self-supervised ResNet-50 (MoCo), with trajectory examples provided in the supplementary (F.0.1).

**PGDD generalizes across architectures and layers.** PGDD demonstrates remarkable generalizability across diverse network architectures and representational levels. Figure 3 shows PGDD generation results across: (A) different robustness parameters, (B) $L_\infty$ vs $L_2$ adversarial training norms (Madry et al., 2018), (C) multiple CNN architectures including VGG16-BN (Simonyan & Zisserman, 2014), ResNet-18, ResNet-50, and the wider ResNeXt-50-32x4d (Xie et al., 2017), (D) Vision Transformer architecture (ViT-S-CvSt, (Singh et al., 2023)) , and (E) different internal layers of ResNet-50. Starting from identical noise initialization, PGDD consistently produces semantically coherent, class-consistent patterns across all tested conditions. The diversity of generated semantic categories—from animals and vehicles to natural scenes—demonstrates that the implicit denoising structure $J^T J$ scales effectively across ImageNet's 1000 learned categories and generalizes beyond specific architectural choices. Importantly, the consistent generation quality across architectures supports our theoretical claim that spectral concentration is a fundamental property of robust training rather than an architecture-specific artifact.

**PGDD trajectories seem to converge to stable patterns.** To understand PGDD's behavior systematically, we conduct comprehensive experiments across multiple robust classifiers trained with different adversarial perturbation budgets ($\epsilon$), training epochs, and initialization seeds. For each model, we run PGDD starting from 100 different noise seeds with 10 different inference seeds, generating 1000 trajectories per model. The results reveal consistent convergence properties (Figure 4): despite diverse starting conditions, trajectories converge to stable class predictions with high confidence. Multiple runs frequently arrive at the same predicted class, suggesting that PGDD reliably accesses the most prominent modes of the implicit generative model. The generated patterns are distinct from real training digits, appearing more like internal prototypical templates that serve as attractors for PGDD trajectories. Rather than reproducing memorized training examples, the implicit denoiser $J^\top J$ reveals canonical digit representations that guide the convergence dynamics, although further work is needed to characterize the formation of these stable patterns .

**Generated patterns reveal distinct prototypical features.** To characterize the generated patterns against an equivalent number of samples from training data, we employ t-SNE visualization (van der Maaten & Hinton, 2008) (Figure 5) to investigate their similarity to real data and examine their distinct characteristics. As expected, the generated patterns occupy regions very distinct from real training data, confirming that PGDD does not simply reproduce memorized examples. However, when samples are colored by class (predicted class for generated samples), we observe distinct clusters of similarly recognized patterns that remain stable across different t-SNE random initialization seeds (See also Supplementary F). This consistent clustering suggests that these generated patterns may represent the stable discriminative features the robust model has learned during training - canonical representations that serve as attractors in the implicit generative space rather than copies of specific training instances. The separation between generated and real data, combined with the coherent and reproducible class-based clustering of generated samples, provides evidence that robust classifiers encode prototypical feature representations that extend beyond the original training distribution.

## 5 LIMITATIONS

While our results provide strong evidence for the implicit denoiser hypothesis, several limitations suggest directions for future work. Our detailed spectral analysis of $J^\top J$ eigenstructure is currently limited to relatively simple architectures and small dataset (MNIST) due to computational tractability. Also, the space of possible generative outputs accessible through PGDD is vast, determined by the complex interaction between input noise patterns, algorithm hyperparameters, and the network's

learned priors. Our experiments represent only a small fraction of this combinatorial space. The rich diversity of patterns achievable through different parameter configurations suggests that systematic exploration of this landscape could reveal much deeper insights into the structure of implicit generative models. This limitation also presents an opportunity: PGDD exhibits natural convergence properties, with trajectories consistently arriving at stable attractors in the learned prior distribution. With appropriate hyperparameter tuning strategies, the algorithm could automatically navigate to the nearest meaningful attractor without manual parameter selection. Developing principled methods for adaptive hyperparameter adjustment based on convergence dynamics would significantly improve the method's practical applicability across different architectures and datasets. Such automation would also enable more systematic exploration of the vast space of implicit priors encoded by robust classifiers.

Finally, while we demonstrate correlations between spectral concentration and denoising capability, establishing direct causal relationships would strengthen our theoretical framework. Controlled experiments that systematically perturb the spectral properties of $\mathbf{J}^\top \mathbf{J}$ through targeted interventions during training or post-hoc modifications, and measure the resulting impact on generative quality, would provide more definitive evidence for the mechanistic role of eigenvalue structure. Such experiments would help distinguish between correlation and causation in the relationship between robustness training and generative capabilities, potentially revealing new ways to enhance implicit denoising through architectural or training modifications.

## 6 CONCLUSION

This work provides evidence for a connection between adversarial robustness and generative modeling by demonstrating that robust classifiers appear to contain implicit denoising structure encoded in their Jacobian operators. Our theoretical framework suggests that the Gram operator $\mathbf{J}^\top \mathbf{J}$ can function as an implicit denoiser through the low-rank spectral structure induced by adversarial training. This mathematical insight offers a potential bridge between discriminative robustness and generative inference within a unified framework. Our empirical validation across datasets and architectures confirms the theoretical predictions. Robust classifiers exhibit stronger denoising capabilities compared to standard networks, with spectral properties that enable selective amplification of discriminative directions while suppressing noise. Visual analysis reveals that applying $\mathbf{J}^\top \mathbf{J}$ to random perturbations produces class-consistent structure in robust models but preserves random patterns in standard classifiers.

The Prior-Guided Drift Diffusion (PGDD) algorithm translates these theoretical insights into practical generation capabilities. PGDD leverages implicit denoising structure through simple inference objectives rather than explicit Jacobian computation, enabling coherent image synthesis from noise without requiring generative training or architectural modifications. Large-scale trajectory analysis demonstrates consistent convergence properties across diverse initialization conditions, indicating that PGDD reliably accesses meaningful statistical priors rather than exploiting spurious patterns. The extension to standard networks through sPGDD reveals that implicit generative structure exists beyond robustly trained classifiers, though with reduced fidelity. Notably, both approaches—adversarial training (which smooths the loss landscape during training) and sPGDD (which smooths gradients during inference)—demonstrate that regularization techniques can provide access to these implicit generative structures, suggesting that smoothing mechanisms may be a general principle for exposing the dual discriminative-generative nature of neural networks.

The finding that PGDD-generated samples occupy distinct manifolds from training data while maintaining class consistency demonstrates that robust classifiers encode implicit structural knowledge, though their current generative capacity remains limited. These findings suggest potential applications in interpretability and explainability by enabling access to the learned priors embedded within discriminative networks. The observed generative properties indicate that robust training may encode richer representations of data distributions than previously recognized, though further investigation is needed to characterize these representations fully. This work provides a foundation for exploring connections between robustness and generation. The implicit structure revealed here may offer new insights into why these joint objectives remain challenging and potentially suggest alternative approaches for leveraging the dual nature of these learned representations.

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

## A  APPENDIX

## B  SPECTRAL EXPERIMENTS SETUP

All MNIST and spectral experiments in this paper were conducted using the reference CNN and training code from the NeurIPS 2018 tutorial *Adversarial Robustness: Theory and Practice* by Zico Kolter and Aleksander Madry Kolter & Madry (2018). We made minimal adaptations (e.g. adding an $\ell_2$-PGD attack and logging spectral metrics).

**Architecture.**  The CNN used in all MNIST experiments has the following structure:

- Conv2d(1→32, kernel=3, padding=1), ReLU
- Conv2d(32→32, kernel=3, padding=1, stride=2), ReLU
- Conv2d(32→64, kernel=3, padding=1), ReLU
- Conv2d(64→64, kernel=3, padding=1, stride=2), ReLU
- Flatten
- Linear(7·7·64 → 100), ReLU
- Linear(100 → 10)

$\ell_2$**-PGD attack (training and evaluation).**  All adversarial training and evaluation use $\ell_2$-PGD with: $\epsilon = 1.5$, step size $\alpha = 0.2$, 20 iterations, projection onto the $\ell_2$-ball per example, and clamping to $[0, 1]$. Random start was disabled unless otherwise noted.

**Spectral metrics.**  For each checkpoint we compute Jacobian-based spectral quantities on held-out data, including the energy ratio derived from $Q = J^\top J$ (expectations over multiple random $\xi$), and power-law exponents fit to the spectrum. These metrics are used to map $(\varepsilon, \text{epoch})$ phase diagrams.

**Accuracies.**  Final clean and robust accuracies will be reported in Table 2.

Table 2: MNIST CNNs for Figure 1 and Table 1. Robust accuracy measured under $\ell_2$-PGD ($\epsilon = 1.5$, 20 steps).

| Model | Clean Acc. | Robust Acc. |
|---|---|---|
| Clean CNN | 0.98 | 0.02 |
| Robust CNN | 0.98 | 0.47 |

## C  SUPPLEMENTARY: IMPLICIT DENOISER PROBES

To produce the quantitative results reported in Table 1 of the main text, we ran three complementary probes designed to test the hypothesis that $J^\top J$ acts as an implicit denoiser in robust classifiers. All code is adapted from our MNIST experimental framework (see Section S1). Below we summarize the key implementation details.

**Energy ratios.**  For a held-out test input $x$, we repeatedly sample isotropic Gaussian noise $\epsilon \sim \mathcal{N}(0, \sigma^2 I)$ and compute

$$r = \frac{\|J^\top J \epsilon\|}{\|\epsilon\|}.$$

We report the mean of $r$ across $n = 128$ draws, with $\sigma = 0.25$. Lower values indicate stronger denoising by suppression of noise directions.

**Eigenvalue decay.**  We estimate the leading eigenvalues of $J^\top J$ at test points using subspace iteration with Gram–Schmidt re-orthogonalization. Starting from $k$ random vectors (we use $k = 8$), we repeatedly apply $J^\top J$ and orthogonalize, then compute Rayleigh quotients $\lambda_i = \langle v_i, J^\top J v_i \rangle$. We report the sorted eigenvalues and the $\lambda_1/\lambda_2$ ratio to capture spectral dominance.

**Visual residuals.** For qualitative inspection, we sample a random $\epsilon$ at fixed scale $\sigma = 0.35$ and display both the raw input noise and its transformation $J^\top J\epsilon$. This illustrates that robust models suppress noise and emphasize structured residuals, consistent with implicit denoising.

## C.1 SPECTRAL PROPERTIES FOR INCREASING ROBUSTNESS ACROSS MNIST, CIFAR10 AND IMAGENET

We analyze the spectral properties of TinyResNet models trained on MNIST with varying levels of adversarial robustness. The TinyResNet architecture is a lightweight ResNet variant implemented using the Hugging Face transformers library, consisting of two stages with basic residual blocks. Specifically, the architecture uses depths $[2, 2]$ (two blocks per stage) and hidden channel sizes $[32, 64]$ (32 channels in the first stage, 64 in the second), with skip connections enabling gradient flow through the network. Models are trained using L2 PGD adversarial training with training perturbation budgets $\varepsilon_{\text{train}} \in \{0.00, 0.25, 0.50, 0.75, 1.00\}$.

Figure S4 visualizes how the Jacobian transpose-Jacobian operator $J^T J$ transforms random input noise $\varepsilon$ for models with increasing robustness. The computation proceeds as follows: (1) a single random noise pattern $\varepsilon$ is generated and normalized to $[0, 1]$; (2) for each model, we compute $J^T J\varepsilon$ at a test image $x$ using automatic differentiation, where $J$ is the Jacobian of the model's logit output with respect to the input; (3) the transformation $J^T J\varepsilon$ is computed implicitly via Jacobian-vector products (JVP) followed by vector-Jacobian products (VJP) to avoid explicitly constructing the full Jacobian matrix.

The figure demonstrates a key spectral property: as the training perturbation budget $\varepsilon_{\text{train}}$ increases, the operator $J^T J$ becomes more selective, extracting clearer patterns from random noise. For the standard model ($\varepsilon_{\text{train}} = 0.00$), the transformed residual remains largely noisy, while robust models ($\varepsilon_{\text{train}} \geq 0.50$) reveal distinct digit-like structures, indicating that $J^T J$ acts as a pattern extraction operator whose selectivity increases with adversarial training. This visualization provides an intuitive understanding of how robust models develop more structured input-output relationships, with the Jacobian transpose-Jacobian operator effectively filtering noise to reveal the underlying data manifold structure.

Table 3 summarizes the quantitative spectral properties across all models. As robustness increases, we observe several key trends: (1) the trace $\text{tr}(J^T J)$ decreases monotonically, indicating reduced overall sensitivity; (2) the energy ratio $\|J^T J\varepsilon\|/\|\varepsilon\|$ decreases, showing that the operator becomes more selective; (3) the top eigenvalues $\lambda_1$ and $\lambda_2$ decrease, with their ratio $\lambda_1/\lambda_2$ initially increasing then stabilizing, suggesting a shift in the eigenspectrum structure; (4) the R² values for the $\Delta_{\text{robust}} \approx \alpha \cdot \sigma^2$ relationship improve with robustness, indicating better predictability of robustness gaps. These quantitative measures complement the visual analysis, confirming that adversarial training fundamentally alters the spectral properties of the model's input-output mapping.

Table 3: Spectral properties summary for TinyResNet models trained with varying adversarial perturbation budgets $\varepsilon_{\text{train}}$ on MNIST. Clean Acc and Robust Acc denote clean and robust test accuracies, respectively. The trace $\text{tr}(J^T J)$ measures overall operator magnitude, while $\|J^T J\varepsilon\|/\|\varepsilon\|$ is the median energy ratio. The eigenvalues $\lambda_1$ and $\lambda_2$ are the top two eigenvalues of $J^T J$.

| Model | Clean Acc | Robust Acc | $\text{tr}(J^T J)$ | $\frac{\|J^T J\varepsilon\|}{\|\varepsilon\|}$ | $\lambda_1$ | $\lambda_2$ | $\frac{\lambda_1}{\lambda_2}$ |
|---|---|---|---|---|---|---|---|
| $\varepsilon = 0.00$ | 0.9844 | 0.9844 | 127.1337 | 1.1854 | 33.5095 | 19.6171 | 1.71 |
| $\varepsilon = 0.25$ | 0.9882 | 0.9770 | 58.8217 | 0.6695 | 19.0332 | 10.2712 | 1.85 |
| $\varepsilon = 0.50$ | 0.9898 | 0.9678 | 41.3736 | 0.3697 | 10.8113 | 5.6020 | 1.93 |
| $\varepsilon = 0.75$ | 0.9891 | 0.9533 | 31.7437 | 0.3725 | 11.6364 | 3.5081 | 3.32 |
| $\varepsilon = 1.00$ | 0.9911 | 0.9410 | 23.8401 | 0.2457 | 7.2376 | 3.4450 | 2.10 |

We extend this spectral analysis to more complex datasets and architectures to validate the generality of our findings. Figure S2 demonstrates the same spectral concentration phenomenon on CIFAR-10 using ResNet-50 models trained with varying adversarial perturbation budgets $\varepsilon_{\text{train}} \in \{0.00, 0.25, 0.50, 1.00\}$. The top row shows five different random noise samples $\varepsilon$, while subsequent rows display the visual residuals $J^T J\varepsilon$ for each model. Consistent with our MNIST findings, standard CIFAR-10 classifiers ($\varepsilon = 0.00$) preserve much of the random structure in the

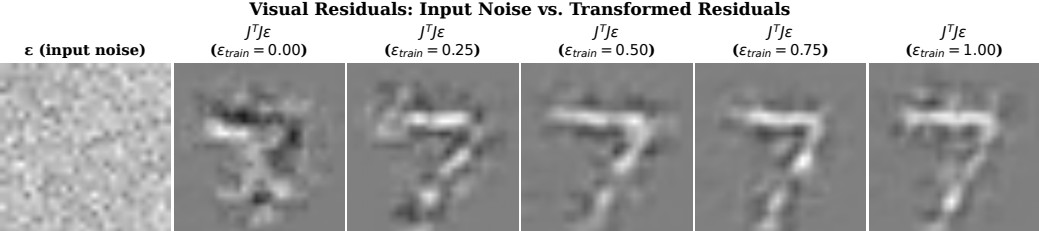

Figure S1: Visual residuals showing the transformation of random input noise $\varepsilon$ by the operator $J^T J$ for TinyResNet models trained with increasing adversarial perturbation budgets $\varepsilon_{\text{train}}$. The leftmost panel shows the input noise pattern, while subsequent panels display $J^T J \varepsilon$ for models trained with $\varepsilon_{\text{train}} \in \{0.00, 0.25, 0.50, 0.75, 1.00\}$. As robustness increases, the operator becomes more selective, extracting clearer digit-like patterns from the noise, demonstrating how adversarial training shapes the spectral properties of the model's input-output mapping.

noise, producing residuals that closely resemble the input patterns. In contrast, robust classifiers show progressive noise suppression and structure extraction as training robustness increases. For $\varepsilon = 1.00$, the operator $J^T J$ transforms random noise into coherent spatial patterns with clear edge structures and color organization, demonstrating that spectral concentration enables semantic pattern extraction even in complex natural image datasets.

Figure S3 provides validation at the largest scale, showing spectral residuals for ResNet-50 models trained on ImageNet with perturbation budgets $\varepsilon_{\text{train}} \in \{0.00, 3.00, 5.00\}$. Despite the significantly higher complexity of ImageNet (1000 classes, 224×224 resolution), the fundamental spectral properties remain consistent. Standard ImageNet classifiers produce residuals that largely preserve the random structure of input noise, while robust models ($\varepsilon = 3.00, 5.00$) generate structured patterns with recognizable spatial organization and semantic coherence. The residuals reveal edge-like structures, texture patterns, and spatial arrangements that suggest the network has learned to extract meaningful visual features from random inputs. This cross-dataset consistency—from simple MNIST digits to complex natural images—provides strong empirical support for our theoretical framework.

The quantitative trends observed in MNIST (Table 3) are replicated across all three datasets, with energy ratios $\|J^T J \varepsilon\| / \|\varepsilon\|$ decreasing monotonically with increasing robustness and trace values $\text{tr}(J^T J)$ showing systematic reduction. This systematic behavior across datasets of varying complexity and different network architectures suggests that spectral concentration is a fundamental consequence of adversarial training rather than a dataset-specific artifact. The visual evidence from Figures S4, S2, and S3 collectively demonstrates that the Jacobian transpose-Jacobian operator $J^T J$ transitions from a noise-preserving transformation in standard networks to a semantic pattern extractor in robust networks, providing the mechanistic foundation for the generative capabilities we observe through PGDD.

**ImageNet spectral summary.** Finally, our quantitative analysis on ImageNet (Table 4), together with the normalized eigenvalue decay shown in Figure S4, confirms that the same spectral trends observed on MNIST and CIFAR-10 persist at large scale. The standard ImageNet model exhibits extremely high energy ratios and large trace values, indicating strong sensitivity to random perturbations and a broadly distributed spectrum. In contrast, robust ImageNet models show two hallmarks of implicit denoiser structure: (1) *aggressive noise suppression*, with gains reduced by more than five orders of magnitude relative to the standard model, and (2) *spectral concentration*, reflected in both the steep normalized eigenvalue decay curves and the consistently low soft-rank values ($\sim$4–5) despite the high dimensionality of the input space. These results demonstrate that adversarial training induces a low-dimensional, highly structured spectral geometry even in high-resolution, 1000-class ImageNet models, reinforcing our central claim that robustness universally drives $J^\top J$ toward a denoising operator across datasets and scales.

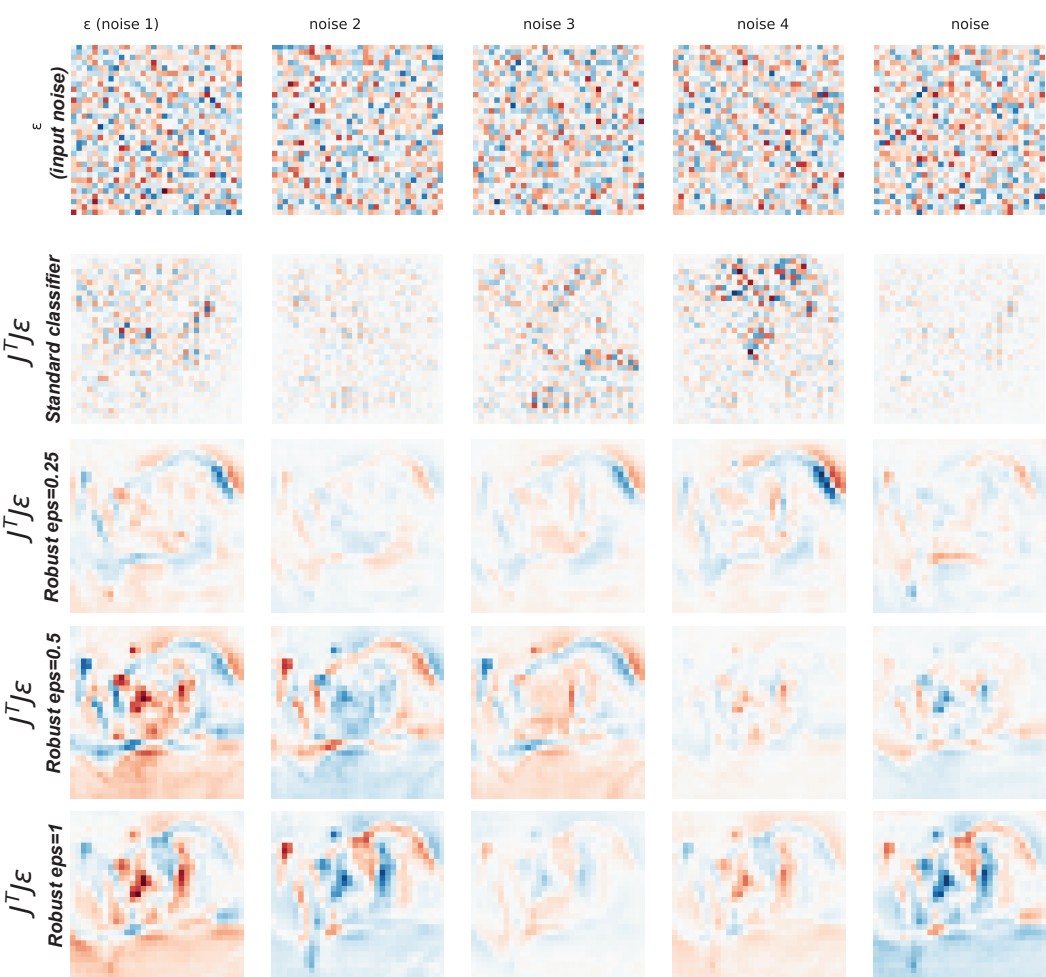

Figure S2: Visual residuals showing the action of $\mathbf{J}^T\mathbf{J}$ on random noise for CIFAR10 ResNet50 models with varying robustness. **Top row:** Five different input noise samples ($\varepsilon$). **Subsequent rows:** Visual residuals ($\mathbf{J}^T\mathbf{J}\varepsilon$) for a standard classifier ($\varepsilon = 0.00$) and robust classifiers with increasing adversarial training strength ($\varepsilon = 0.25, 0.50, 1.00$). All visualizations use symmetric normalization (red-blue diverging colormap) with per-model scaling.

Table 4: **ImageNet spectral statistics.** Spectral properties of ImageNet ResNet-50 models under varying $\ell_2$ adversarial robustness. Columns report noise suppression (Gain), Hutchinson trace $\text{tr}(J^\top J)$, leading eigenvalues, normalized eigenvalue ratio, spectral tail, entropy-based soft-rank, and PGDD denoising performance ($\Delta$PSNR).

| Model | Gain | $\text{tr}(J^\top J)$ | $\lambda_1$ | $\lambda_2$ | $\lambda_1/\lambda_2$ | tail@$k$ | soft-rank | $\Delta$PSNR (dB) |
|---|---|---|---|---|---|---|---|---|
| $\epsilon = 0.00$ (Std.) | 116.28 | 152,042.0 | 34,815.5 | 18,709.4 | 1.86 | 4,538.0 | 4.77 | -2.80 |
| $\epsilon = 1.00$ | 0.0044 | 7.37 | 1.82 | 0.82 | 2.22 | 0.11 | 4.04 | -0.80 |
| $\epsilon = 2.00$ | 0.0012 | 1.28 | 0.45 | 0.18 | 2.51 | 0.03 | 4.13 | 1.48 |
| $\epsilon = 3.00$ | 0.0001 | 0.24 | 0.14 | 0.06 | 2.26 | 0.01 | 4.33 | 1.21 |
| $\epsilon = 5.00$ | 0.0002 | 0.16 | 0.08 | 0.03 | 2.34 | 0.01 | 4.42 | 0.83 |

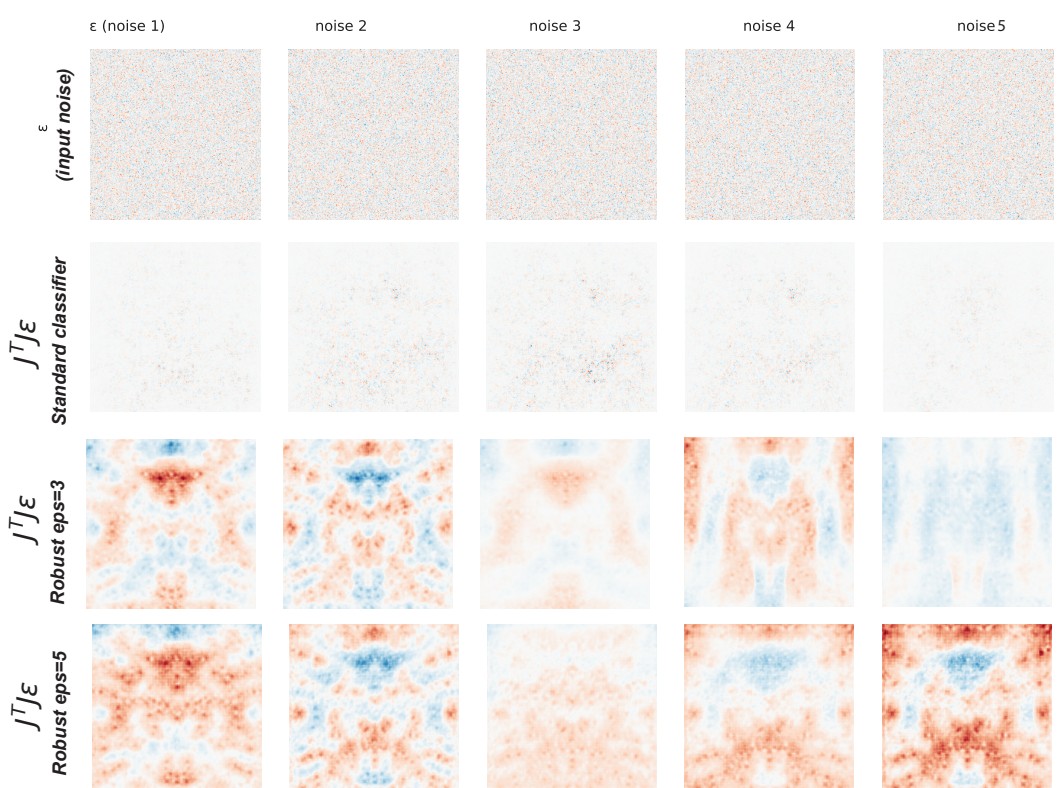

Figure S3: Visual residuals showing the action of $\mathbf{J}^T\mathbf{J}$ on random noise for ImageNet ResNet50 models with varying robustness. **Top row:** Five different input noise samples ($\varepsilon$). **Subsequent rows:** Visual residuals ($\mathbf{J}^T\mathbf{J}\varepsilon$) for a standard classifier ($\varepsilon = 0.00$) and robust classifiers with increasing adversarial training strength ($\varepsilon = 3.00, 5.00$). All visualizations use symmetric normalization (red-blue diverging colormap) with per-model scaling.

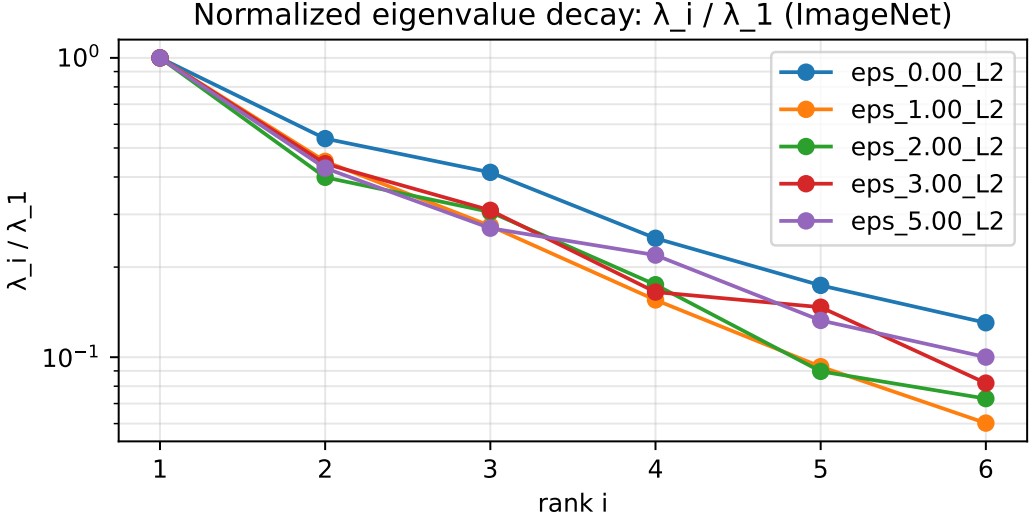

Figure S4: Normalized eigenvalue decay. Top-$k$ normalized eigenvalues $\lambda_i/\lambda_1$ of $J^\top J$ for standard and robust ImageNet classifiers. (eps =0 shows a standard resnet50, all trained on ImageNet)

## C.2 PRIOR-GUIDED DRIFT DIFFUSION: ALGORITHM, INTUITION, AND THEORY

We present the detailed algorithm for Prior-Guided Drift Diffusion (PGDD), together with the underlying intuition and theoretical justification for how PGDD grants access to the learned priors of a network. The method is designed to be both conceptually transparent and practically simple, offering a principled way to leverage the implicit generative structure in networks which were not explicitly trained for pattern generation (notably classifiers).

---

**Algorithm 1** Prior-Guided Drift Diffusion Objective

---

1: **Input:** Image $\mathbf{x}_{\text{input}}$, model $f$, target layer $\ell$, constraint $\epsilon$, step size $\alpha$, noise ratio $\sigma$, iterations $T$
2: **Output:** Refined representations $\{\mathbf{x}_t\}_{t=0}^{T}$
3: // Step 1: Feedforward pass
4: $\mathbf{x}_0 \leftarrow \text{normalize}(\mathbf{x}_{\text{input}})$
5: $f_\ell \leftarrow \text{extract\_layers}(f, \ell)$ {Extract model up to layer $\ell$}
6: $\mathbf{x}_{\text{noisy}} \leftarrow \mathbf{x}_0 + \sigma \cdot \mathcal{N}(0, I)$
7: $r_{\text{anti-target}} \leftarrow f_\ell(\mathbf{x}_{\text{noisy}})$ {Generate noisy reference representation}
8: **for** $t = 1$ to $T$ **do**
9:     // Step 2: Inference objective selection
10:     anti-target $\leftarrow r_{\text{anti-target}}$ {Use noisy reference as target}
11:     // Step 3: Feedback error propagation
12:     $h_t \leftarrow f_\ell(\mathbf{x}_{t-1})$ {Forward pass through target layers}
13:     $L_t \leftarrow \|h_t - r_{\text{anti-target}}\|^2$ {MSE loss in representation space}
14:     $g_t \leftarrow \nabla_{\mathbf{x}_{t-1}} L_t$ {Gradient via feedback pathways}
15:     // Step 4: Constrained activation update
16:     $\tilde{g}_t \leftarrow \alpha \cdot g_t / (\|g_t\| + 1\text{e-}10)$ {Normalize gradient}
17:     $\eta_t \leftarrow \text{diffusion\_noise\_ratio} \cdot \mathcal{N}(0, I)$ {Add stochastic noise}
18:     $\mathbf{x}'_t \leftarrow \mathbf{x}_{t-1} + \tilde{g}_t + \eta_t$ {Move away from representation of noisy input}
19:     $\mathbf{x}_t \leftarrow \text{project}(\mathbf{x}'_t, \mathbf{x}_0, \epsilon)$ {Enforce $\|\mathbf{x}_t - \mathbf{x}_0\|_\infty \le \epsilon$}
20: **end for**
21: **Return** $\{\mathbf{x}_t\}_{t=0}^{T}$ =0

---

## C.3 PGDD PARAMETERS FOR MAIN TEXT FIGURES

Table 5: PGDD parameter settings for main test figures (ImageNet ResNet-50, $L_2$ adversarially trained).

| Figure | Model | Loss Inference | Loss Function | Drift Noise Ratio | Diffusion Noise Ratio | $n_{\text{itr}}$ | $\epsilon_{infer}$ | Step Size |
|---|---|---|---|---|---|---|---|---|
| 2 | ResNet-50, $L_2$, $\epsilon = 3$ | PGDD | MSE | 0.2 | 0.01 | 1001 | 40 | 1 |
| 2 | ResNet-50, $L_2$, $\epsilon = 5$ | PGDD | MSE | 0.5 | 0.03 | 1001 | 40 | 1 |
| 3 | vgg16-bn, $L_2$, $\epsilon = 3$ | PGDD | MSE | 0.2 | 0.01 | 1001 | 40 | 1 |
| 3 | resnext50-4dx32, $L_2$, $\epsilon = 3$ | PGDD | MSE | 0.2 | 0.01 | 1001 | 40 | 1 |

## D  FID SCORE FOR EVALUATION OF GENERATED PATTERNS

While Fréchet Inception Distance (FID) (Heusel et al., 2017) has become the standard metric for evaluating generative model quality, its application to MNIST presents fundamental limitations. The Inception network used in standard FID computation is trained on ImageNet, which contains natural images with fundamentally different statistical properties than handwritten digits. This domain mismatch makes Inception-based FID scores unreliable for MNIST evaluation, as the feature representations may not capture the relevant structure for digit classification. Additionally, for ImageNet-scale generation, computing FID scores would require generating thousands of samples to obtain statistically meaningful comparisons, which is computationally prohibitive for our large-scale trajectory analysis.

To address these limitations, we introduce a FID-like score that uses domain-appropriate feature extractors. For MNIST, we employ TinyResNet models trained on MNIST as feature extractors, computing the Fréchet distance between feature distributions of real test images and PGDD-generated samples. This approach provides a principled evaluation metric that aligns with the domain of interest while maintaining the statistical rigor of the original FID framework.

Our FID-like score computation follows the standard FID procedure but replaces Inception with TinyResNet feature extractors. Specifically, we:

1. Extract penultimate layer features from both real MNIST test images and PGDD-generated samples using a reference TinyResNet model

2. Compute the mean $\boldsymbol{\mu}$ and covariance $\boldsymbol{\Sigma}$ of the feature distributions for both sets

3. Calculate the Fréchet distance between the two multivariate Gaussian distributions:
$$\text{FID} = \|\boldsymbol{\mu}_r - \boldsymbol{\mu}_g\|^2 + \text{tr}(\boldsymbol{\Sigma}_r + \boldsymbol{\Sigma}_g - 2(\boldsymbol{\Sigma}_r \boldsymbol{\Sigma}_g)^{1/2}), \tag{11}$$
where subscripts $r$ and $g$ denote real and generated distributions, respectively.

The generated patterns used for FID-like score computation were obtained through systematic parameter sweeps across multiple dimensions. For each robust TinyResNet model (trained with different adversarial perturbation budgets $\varepsilon_{\text{train}} \in \{0.25, 1.00\}$), we generated samples by:

- **Input noise seed sweep:** 100 different input noise seeds (ranging from 1000-1100) to vary the initial starting conditions

- **Inference seed sweep:** 10 different inference seeds (ranging from 2000-2009) to vary the stochastic components of PGDD

- **PGDD parameter configuration:** Fixed PGDD parameters (step size=0.5, diffusion noise ratio=0.005, initial inference noise ratio=0.5, $\epsilon_{\text{infer}} = 2.0$, iterations=50) consistent with the main experimental setup

This systematic sweep produced 1000 trajectories per model configuration, from which we extracted the final iteration images (iteration 49) for FID-like score computation. The diversity of initialization conditions ensures that our evaluation captures the full range of generative patterns accessible through PGDD rather than cherry-picking specific examples.

We evaluate FID-like scores using three different reference models to understand how the choice of feature extractor affects the assessment of generative quality:

- **Clean model** ($\varepsilon = 0.00$)**:** Standard TinyResNet trained without adversarial perturbations

- **Moderately robust model** ($\varepsilon = 0.25$)**:** TinyResNet trained with $\ell_2$ adversarial training at $\varepsilon = 0.25$

- **Highly robust model** ($\varepsilon = 1.00$)**:** TinyResNet trained with $\ell_2$ adversarial training at $\varepsilon = 1.00$

This multi-reference evaluation reveals how the feature space geometry varies with robustness training and provides insights into which models' priors are best captured by PGDD generation.

Table 6 presents FID-like scores computed across different combinations of reference models and generated image sources. Several key observations emerge:

Table 6: FID-like scores for PGDD-generated MNIST samples using different TinyResNet reference models. Lower scores indicate greater similarity between generated and real image feature distributions. Scores computed using 10,000 real MNIST test images and 995-1000 generated samples per configuration.

| Reference Model | Generated from $\varepsilon = 0.25$ | Generated from $\varepsilon = 1.00$ |
|---|---|---|
| $\varepsilon = 0.00$ (clean) | 119.98 | 98.73 |
| $\varepsilon = 0.25$ | 101.96 | 97.86 |
| $\varepsilon = 1.00$ | 88.69 | 133.11 |

The choice of reference model significantly affects FID-like scores, reflecting fundamental differences in feature space geometry. When using the clean model ($\varepsilon = 0.00$) as reference, images generated from the highly robust model ($\varepsilon = 1.00$) achieve lower FID scores (98.73) than those from the moderately robust model ($\varepsilon = 0.25$, score 119.98). This suggests that highly robust models produce patterns that are more aligned with the clean model's feature space, despite being trained with stronger adversarial constraints. Interestingly, when using the same model as both generator and reference, the results are not uniformly favorable. For the $\varepsilon = 0.25$ reference model, its own generated images achieve a score of 101.96, while $\varepsilon = 1.00$ generated images score slightly better at 97.86. Conversely, when using the $\varepsilon = 1.00$ model as reference, $\varepsilon = 0.25$ generated images achieve the best score (88.69), while its own generated images score worse (133.11). This asymmetry suggests that the feature representations learned by different robustness levels are not simply nested subspaces but rather exhibit complex geometric relationships.

The FID-like scores demonstrate that PGDD successfully generates patterns that occupy meaningful regions of the feature space, with scores in the range of 88-134. While these values are higher than typical FID scores for high-quality generative models (which often achieve scores below 10), this is expected given that: (1) PGDD operates without explicit generative training, (2) the feature space dimensionality (10 dimensions for TinyResNet penultimate layer) is much smaller than Inception's 2048-dimensional space, and (3) the generated patterns represent prototypical attractors rather than direct reproductions of training data. The consistent convergence to stable patterns (as evidenced by the trajectory analysis in Section 4) combined with these FID-like scores provides quantitative evidence that robust classifiers encode meaningful generative priors accessible through PGDD.

We note that direct comparison to standard Inception-based FID scores is not meaningful due to the domain mismatch discussed above. However, our FID-like scores serve as a principled evaluation metric that: (1) uses domain-appropriate feature extractors, (2) maintains the statistical rigor of the Fréchet distance framework, and (3) enables quantitative assessment of generative quality for MNIST without requiring thousands of ImageNet-scale generations. This approach provides a practical solution for evaluating implicit generative capabilities in robust classifiers while remaining computationally tractable for systematic parameter sweeps.

# E  ANALYSIS OF GENERATED DIGIT DISTRIBUTION AND PER-CLASS ACCURACY

To understand how the implicit generative priors encoded by robust classifiers relate to their discriminative performance, we conducted a comprehensive analysis comparing the distribution of digits generated by PGDD with the per-class classification accuracy of the same models on the test set. This analysis provides insights into whether models generate digits in proportion to their classification performance, revealing potential biases in the learned representations.

We evaluated multiple ResNetTiny models trained on MNIST with different adversarial perturbation budgets ($\epsilon \in \{0.25, 0.50, 0.75, 1.00\}$) and at various training epochs (excluding epoch 5 to focus on more mature models). For each model checkpoint, we collected all generated images produced by PGDD across multiple inference runs with varying input noise seeds and inference parameters. We extracted the final predicted digit class for each generated image (the class predicted by the model at the final iteration of PGDD) and computed the distribution of generated digits across all 10 classes. This distribution represents what the model's implicit generative prior considers to be the most natural or prototypical digit representations. For the same model checkpoints, we evaluated classification performance on the MNIST test set, computing both clean accuracy and adversarial robust accuracy for each digit class (0-9). This provides a measure of how well the model discriminates between different digit classes, allowing us to assess whether generation frequency correlates with classification performance.

Our analysis reveals that the distribution of generated digits is highly non-uniform across all models, with certain digits (notably digits 4 and 8) being generated significantly more frequently than others. This non-uniformity persists across different training epochs and perturbation budgets, suggesting that robust training induces consistent biases in the learned generative priors. Interestingly, the frequency with which digits are generated does not directly correlate with per-class classification accuracy. While models achieve relatively uniform clean accuracy across digit classes (typically 96-99%), the generated digit distribution shows substantial variation, with some digits appearing 5-10 times more frequently than others. This suggests that the implicit generative prior encoded by $\mathbf{J}^\top \mathbf{J}$ reflects structural properties of the learned representations that extend beyond simple classification performance.

The generated digit distribution remains relatively stable across different training epochs for models with the same perturbation budget, indicating that the implicit generative structure emerges early in training and remains consistent. However, we observe subtle variations in the distribution as training progresses, potentially reflecting refinement of the learned priors. Models trained with different adversarial perturbation budgets ($\epsilon$) exhibit distinct generation patterns, with higher $\epsilon$ values generally producing more diverse digit distributions. This observation aligns with our theoretical framework: stronger adversarial training constraints induce more structured $\mathbf{J}^\top \mathbf{J}$ operators, which may access richer generative priors.

These findings suggest that the implicit generative structure in robust classifiers captures statistical regularities that are not directly reflected in classification metrics. The non-uniform generation patterns may reflect the intrinsic difficulty of representing certain digit classes in the learned feature space, or they may indicate that the implicit denoiser $\mathbf{J}^\top \mathbf{J}$ has learned to prioritize certain prototypical digit representations that serve as attractors in the generative space. The stability of generation patterns across training epochs indicates that the implicit generative structure is a fundamental property of the learned representations, rather than an artifact of specific training configurations. This stability supports our theoretical framework, which posits that adversarial training fundamentally alters the spectral structure of the Jacobian operator. The variation in generation patterns with different perturbation budgets provides empirical support for the connection between robustness constraints and generative capabilities, suggesting that stronger adversarial training may enable access to more diverse and structured implicit priors. These findings contribute to our understanding of how discriminative robustness relates to generative modeling, revealing that robust classifiers encode statistical knowledge that extends beyond their primary classification function. The systematic analysis of generation patterns provides a new lens through which to examine the learned representations of robust models, potentially informing future work on interpretability and explainability.

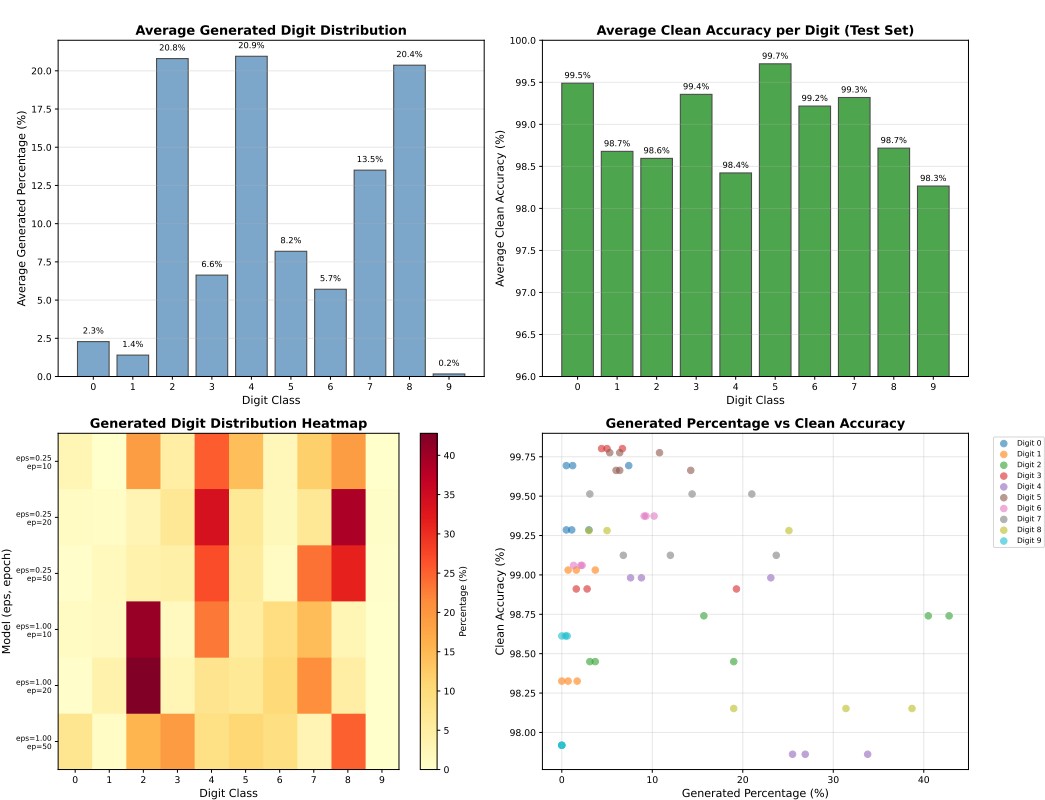

Figure S5: Comparison of generated digit distribution and per-class classification accuracy for ResNetTiny models trained on MNIST with different adversarial perturbation budgets ($\epsilon \in \{0.25, 0.50, 0.75, 1.00\}$) and training epochs. (Top left) Average percentage of generated images classified as each digit class (0-9) across all evaluated models. (Top right) Average clean accuracy per digit class on the test set, with y-axis range 96-100%. (Bottom left) Heatmap of generated digit distribution for each model (rows) and digit class (columns), with color intensity representing the percentage of generated images. (Bottom right) Scatter plot of generated percentage versus clean accuracy for each digit class, with different colors indicating different digits.

## F    NEAREST-NEIGHBOR ANALYSIS ON CIFAR-10

To examine whether PGDD-generated samples reflect instance-level recall of training images, we conducted a nearest-neighbor (NN) analysis comparing each generated image to the full CIFAR-10 training set in both pixel space and logit space. In pixel space, we measured Euclidean distance between the flattened RGB vectors of the generated sample and each training image. To assess similarity in the model's learned representation, we also computed nearest neighbors in logit space, using the penultimate logits of a standard (non-robust) CIFAR-10 ResNet-50; these distances reflect semantic similarity as encoded by the classifier. The closest training images exhibit large pixel-space distances and are visually distinct, differing in background, pose, and object boundaries. Across all generated samples we evaluated (examples shown in Figure S6), we did not observe direct reproduction or interpolation of individual training examples.

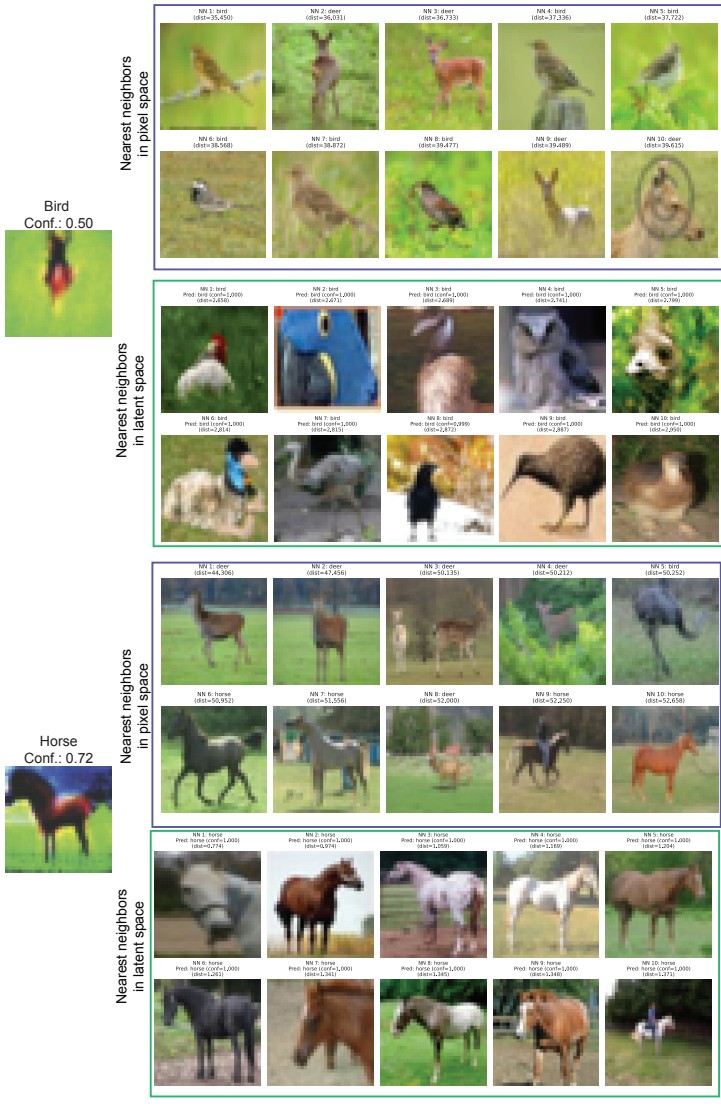

Figure S6: Nearest-neighbor analyses for two PGDD-generated CIFAR-10 samples. Top rows: nearest neighbors in pixel space (Euclidean distance). Bottom rows: nearest neighbors in logit space (Euclidean distance in the model's penultimate logits).

### F.0.1 SPGDD IN STANDARD NETWORKS

Figure S7: **Smooth PGDD (sPGDD) enables generation with standard networks.** sPGDD applied to a standard ResNet-50 (ImageNet, no adversarial training) demonstrates generative capability through gradient smoothing. The method uses multiple independently sampled noise perturbations at each iteration and averages the resulting gradients to reduce variance and emphasize stable prior information embedded in the network. Here, inference starts from a sample of Perline noise.

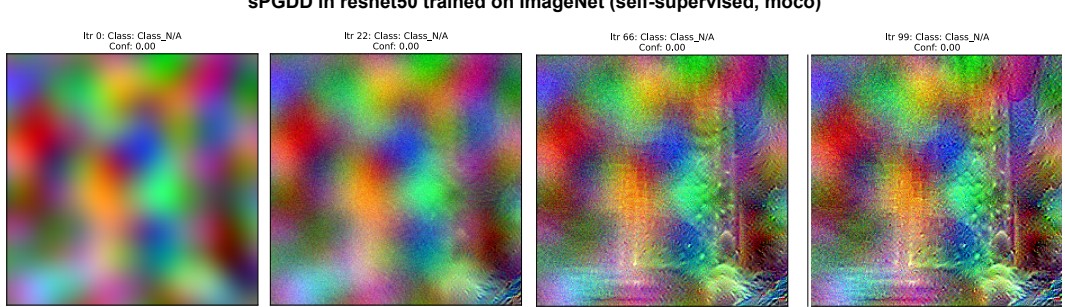

Figure S8: **sPGDD in resnet50 trained with self-supervised objective (moco)**

## F.1 SPGDD PARAMETERS FOR FIGURES S1 AND S2

Table 7: sPGDD parameter settings

| Figure | Model | Loss Inference | Loss Function | Drift Noise Ratio | Diffusion Noise Ratio | $\epsilon_{infer}$ | $\sigma_{smoothing}$ | $n_{smoothing}$ |
|--------|-------|----------------|---------------|-------------------|-----------------------|-------------------|---------------------|-----------------|
| S1 | ResNet-50 | sPGDD | MSE | 0.2 | 0.01 | 40 | 0.1 | 100 |
| S2 | ResNet-50 moco | sPGDD | MSE | 0.5 | 0.01 | 40 | 0.1 | 100 |

