# OpenReview forum: "Implicit Denoiser Structure in Robust Classifiers Explains Generative Capabilities"
_ICLR.cc/2026/Conference — Submitted to ICLR 2026_

### Official Review · Reviewer_BXDv · 2025-10-19

**Soundness:** 2
**Presentation:** 2
**Contribution:** 2
**Rating:** 4
**Confidence:** 3

**Summary:**

This paper investigates the connection between adversarial robustness and generative capabilities. The authors argue that the generative capacity of robust classifiers may originate from an implicit denoising structure within the Gram operator $\mathbf{J}^\top\mathbf{J}$. They propose the PGDD algorithm to leverage this structure for generative tasks. Surprisingly, they extend this mechanism to standard, non-robust models via 'smoothed PGDD.' The authors present numerical experiments to validate their findings.

**Strengths:**

* The authors conduct a careful review of related work, providing a solid foundation and context for their investigation.

* The PGDD algorithm is a conceptually novel and intuitive method. It leverages the hypothesized denoising structure of $J^\top J$ implicitly, bypassing the expensive explicit Jacobian computations.

**Weaknesses:**

* I feel the contributions of the paper are overstated. I find that the two main claims of the paper are not well supported by theoretical or empirical evidence:
  * On the 'Implicit Denoiser' Hypothesis: The paper claims to establish a connection between the Gram operator and an implicit denoising structure. This claim is not substantiated with theoretical rigor, as no formal theorems are presented or proven. The provided numerical analysis only shows a difference in eigenvalue magnitudes between standard and robust models, but does not provide a mechanistic explanation for how this translates to a denoising process that facilitates generation.
  * Second, the authors propose two generating algorithms named PGDD and sPGDD. However, the evidence supporting their generative capabilities is weak and purely qualitative. The authors should have included standard metrics for generative modeling, such as FID or Inception Score, to assess the fidelity and diversity of the generated images.

* The paper is not easy to follow. Some notations, like the Gram operator $\mathbf{J}^\top\mathbf{J}$ and the $R^2$ statistics in Table 1, are not formally defined. (I also guess the energy ratio statistic should be something like $\mathbb{E}\|\mathbf{J}^\top\mathbf{J}\mathbf{\varepsilon}\|/\|\mathbf{\varepsilon}\|$). The description of algorithms like PGDD and sPGDD is too concise, especially for sPGDDs. If this is due to page limits, I suggest the authors shorten the "Conclusion" section or reduce figure sizes.

* Most of the numerical experiments are only conducted with small-scale neural networks on toy datasets like MNIST.

**Questions:**

* I am confused about the specificity of the 'implicit denoising structure'. The paper frames it as a property of robust classifiers that explains their unique generative abilities. However, the proposed sPGDD algorithm demonstrates that a similar generative mechanism can be exploited in standard, non-robust models too. Is this implicit denoising structure a unique characteristic of robust classifiers, or is it a general property of neural networks? If it is not unique, how can it serve as the key explanation for the generative performance of robust models in particular? If it is unique, what allows sPGDD to work on standard models?

---

> ### Author Response · Authors · 2025-11-23
> **Part 1**
>
> Thank you for this detailed and constructive review. We appreciate your recognition that PGDD represents **a conceptually novel and intuitive method**.  We have substantially addressed the raised concerns through theoretical and empirical updates to the manuscript. Specifically, we have added formal theorems with mathematical proofs to establish theoretical rigor, included quantitative generative metrics (FID-like scores for MNIST), expanded our experimental validation to include systematic spectral analysis across MNIST, CIFAR-10, and ImageNet, clarified notation and algorithmic descriptions, and provided detailed theoretical discussion of the relationship between robust and standard networks that addresses your important question about the specificity of implicit denoising structure.
>
> Below, we address each of your specific concerns systematically:
> ### 1. Claims need more support
> >I feel the contributions of the paper are overstated. I find that the two main claims of the paper are not well supported by theoretical or empirical evidence:
> >-   On the 'Implicit Denoiser' Hypothesis: The paper claims to establish a connection between the Gram operator and an implicit denoising structure. This claim is not substantiated with theoretical rigor, as no formal theorems are presented or proven. The provided numerical analysis only shows a difference in eigenvalue magnitudes between standard and robust models, but does not provide a mechanistic explanation for how this translates to a denoising process that facilitates generation.
> >
>
> We appreciate the reviewer's request for greater theoretical rigor. **We have directly addressed this concern by introducing a formal mathematical result: Theorem 1 ("Signal–Noise Decomposition in J^T J").** This theorem rigorously establishes how spectral concentration induced by adversarial training leads to a precise decomposition where J^T J strongly amplifies structured, class-aligned components while suppressing isotropic noise, thereby providing the mechanistic explanation for implicit denoising that the reviewer requested. Additionally, we have expanded our numerical spectral analysis across MNIST, CIFAR-10, and ImageNet with systematic sweeps over robustness values, moving far beyond the limited empirical evidence in the original submission.
>
> **Connecting denoising to generation through established theory:** Regarding the reviewer's question about how denoising properties justify generative capabilities, we emphasize that this connection is fundamental to modern generative modeling. Classical results demonstrate that successful generative models—from denoising autoencoders (Vincent et al., 2008; Bengio et al., 2013) to score-based diffusion models (Song & Ermon, 2019; Ho et al., 2020)—rely on denoising operators. In these frameworks, denoising is the core mechanism that enables generation.
>
> **Theoretical prediction and empirical validation:** Our contribution shows that adversarial training induces precisely the spectral concentration required for J^T J to function as an implicit denoiser. This theoretical framework predicts that robust networks should exhibit strong generative capability through PGDD. Additionally, we hypothesize due to implicit regularization standard neural network have (e.g. https://arxiv.org/abs/2009.11162),  we can arrive at some (albeit weak) generative capability for standard neural network as well.  **Our sPGDD experiments confirm this: even standard networks, when processed with gradient smoothing, can produce faint class-consistent generations exactly of the type predicted by implicit regularization theory—weak, sparse, and highly anisotropic.**
>
> The revised manuscript thus provides the rigorous theoretical framework and comprehensive empirical validation that hopefully addresses the reviewer's concerns about theoretical rigor and mechanistic explanation.

---

> > ### Author Response · Authors · 2025-11-23
> > **Part 2**
> >
> > > Second, the authors propose two generating algorithms named PGDD and sPGDD. However, the evidence supporting their generative capabilities is weak and purely qualitative. The authors should have included standard metrics for generative modeling, such as FID or Inception Score, to assess the fidelity and diversity of the generated images.**
> > **We acknowledge this limitation and have addressed it within the constraints of our research scope and computational resources.**
> >
> > **Added quantitative evaluation for MNIST:** We have implemented a domain-appropriate FID-like score using TinyResNet feature extractors rather than Inception (which is trained on ImageNet and inappropriate for MNIST evaluation). **This analysis required generating substantial numbers of samples (1000+ per configuration) through systematic parameter sweeps across input noise seeds and inference parameters.**
> >
> > For CIFAR-10 and ImageNet, **generating the thousands of samples required for reliable FID computation with our iterative PGDD process (1000+ iterations per sample, 100x for sPGDD) remains computationally challenging for FID scoring at large scale.** Unlike purpose-built generative models that produce samples in single forward passes, our method requires extensive iterative refinement.
> >
> >  We emphasize that **our contribution focuses on understanding the mechanism behind existing generative capabilities** rather than achieving competitive generation quality. As we noted for previous reviewers, **generation quality in our work will never approach models explicitly trained for generation.** Our goal is developing theoretical and empirical tools to explain WHY robust classifiers exhibit (though modest) generative properties through the spectral concentration framework.
> >
> > The quantitative metrics serve to validate our theoretical predictions and enable systematic comparison across robustness levels and architectures, demonstrating that stronger spectral concentration correlates with measurable improvements in generation metrics. This supports our central theoretical claim while acknowledging the inherent limitations of using classifiers for generation tasks.
> >
> >
> > ### 2. Notations definitions
> >
> >   > The paper is not easy to follow. Some notations, like the Gram operator and the statistics in Table 1, are not formally defined. (I also guess the energy ratio statistic should be something like ). The description of algorithms like PGDD and sPGDD is too concise, especially for sPGDDs. If this is due to page limits, I suggest the authors shorten the "Conclusion" section or reduce figure sizes.
> > **Thank you for pointing out these important notation and clarity issues. We have substantially revised the manuscript to address these concerns.**
> >
> > **Formal notation definitions:** We have added a dedicated notation paragraph that formally defines all key terms:
> > - The Gram operator G(x) = J(x)^T J(x) and its eigenvalues λᵢ
> > - The energy ratio statistic ||G(x)ε||₂ / ||ε||₂ for ε ~ N(0,I) (the reviewer correctly guessed this form)
> > - The tail@k metric as λₖ (the k-th largest eigenvalue)
> > - The R² coefficient for the quadratic relationship Δᵣₒᵦᵤₛₜ(σ) ≈ α·σ²
> >
> > **Enhanced Table 1:** We have completely revised Table 1 to be self-contained with:
> > - Clarified column headers and explicit definition of k for tail@k
> > - Expanded caption explaining how each statistic connects to our theoretical predictions in Theorem 1
> > - Clear links between empirical measurements and the signal-noise decomposition framework
> >
> >
> > **PGDD algorithm:** While the detailed pseudocode for PGDD was included in the supplementary materials (Algorithm 1), we acknowledge this may not have been sufficiently prominent. We have now added a concise algorithmic summary in the main text and made the supplementary reference more explicit. **Enhanced sPGDD description:** We agree that the sPGDD description was too conceptual. We have now added a concrete algorithmic description that clarifies the gradient averaging mechanism:
> >
> >
> > ### 3. Numerical experiments on larger dataset
> > > Most of the numerical experiments are only conducted with small-scale neural networks on toy datasets like MNIST.
> > >
> > **We have substantially expanded our experimental scope beyond the original MNIST-focused analysis.** Our revised manuscript now includes: **Systematic spectral analysis across scales:** We now provide comprehensive spectral analysis across MNIST, CIFAR-10, and ImageNet using efficient computational methods (Hutchinson trace estimation, implicit Jacobian-vector products) that avoid the computational intractability of full Jacobian computation at ImageNet scale.  Our key findings—spectral concentration in robust networks, visual residual patterns, and energy ratio reductions—are demonstrated consistently across all three datasets, showing that the phenomenon scales from simple digits to complex natural images.

---

> > > ### Author Response · Authors · 2025-11-23
> > > **Part 3**
> > >
> > > ### Questions
> > > >**Q1**. I am confused about the specificity of the 'implicit denoising structure'. The paper frames it as a property of robust classifiers that explains their unique generative abilities. However, the proposed sPGDD algorithm demonstrates that a similar generative mechanism can be exploited in standard, non-robust models too. Is this implicit denoising structure a unique characteristic of robust classifiers, or is it a general property of neural networks? If it is not unique, how can it serve as the key explanation for the generative performance of robust models in particular? If it is unique, what allows sPGDD to work on standard models?
> > >
> > > Thank you for raising this crucial conceptual question. We agree that the distinction between implicit denoising structure in robust versus standard networks must be made explicit, and **we have clarified this in the revised manuscript.**
> > >
> > > **Our core position is:**
> > > - **Robust classifiers** exhibit a strong, low-rank, class-aligned implicit denoising operator J^T J that enables high-fidelity, class-consistent generation
> > > - **Standard classifiers** exhibit only a weak, noisy, high-rank version of this structure
> > >
> > > **This resolves the apparent tension: sPGDD works on standard models not because they possess the same denoising structure, but because smoothing artificially extracts a very weak approximation of it.**
> > >
> > > **Clarified distinctions (as added to the revised manuscript):**
> > >
> > > **1. Robust classifiers: Strong, intrinsic denoising structure**
> > > Adversarial training shapes J^T J into a low-rank, spectrally concentrated operator aligned with the semantic class manifold. This enables:
> > > - Stable noise suppression (energy ratios ~0.16 vs 6.19 for standard models)
> > > - Clear class-consistent features under J^T J ε (as shown in visual residuals)
> > > - Reliable PGDD generation
> > >
> > > This structure emerges solely from the robustness objective, without any additional smoothing or post-processing.
> > >
> > > **2. Standard classifiers: Weak, high-variance, noisy structure**
> > > Standard networks do contain small pockets of structure—**a byproduct of implicit regularization from SGD (Barrett & Dherin, 2020)**—but this structure is:
> > > - Dispersed across many directions (high-rank Jacobians)
> > > - Drowned in high-variance gradients
> > > - Not aligned cleanly with class manifolds
> > >
> > > Thus, applying J^T J to raw noise produces mostly random patterns, as we show empirically.
> > >
> > > **3. Why sPGDD "works" on standard models**
> > > sPGDD averages gradients over many noise perturbations (n=100), suppressing high-variance components and revealing the tiny amount of coherent structure that does exist. Importantly:
> > > - sPGDD acts as an external denoising mechanism
> > > - PGDD (without smoothing) fails on standard models
> > > - The generative outputs remain low-fidelity, exactly as predicted by our theory
> > >
> > > Thus, sPGDD does not demonstrate that standard models contain robust-like denoisers—only that smoothing can extract a weak approximation of the latent structure that all networks possess to some degree.
> > >
> > > Thank you for this comprehensive review. We believe the above additions directly address your concerns regarding theoretical rigor, notation clarity, algorithmic descriptions, and the specificity of implicit denoising structure. **We have substantially strengthened both our theoretical foundation with formal theorems and our empirical validation across multiple datasets and scales.** If any concerns remain unaddressed or if you would like us to elaborate on any aspects of our theoretical framework or experimental validation, we would greatly welcome your guidance. Are there any other aspects of the work that would benefit from further clarification or analysis? Thank you again for your thoughtful and constructive feedback.

---

> ### Comment · Reviewer_BXDv · 2025-11-27
>
> Thanks for the detailed response. I have carefully reviewed the authors' rebuttal, and I find that it addresses the majority of my concerns. The paper’s logical flow has significantly improved. As a result, I have decided to raise my rating. However, due to my limited familiarity with the literature on adversarial robustness, I will maintain my low confidence score.

---

> > ### Author Response · Authors · 2025-11-27
> >
> > Thank you for taking the time to re-evaluate our manuscript and for raising your rating. We appreciate your careful reading of both the paper and our rebuttal, and we’re grateful that your feedback helped strengthen the manuscript.

---

### Official Review · Reviewer_y7U3 · 2025-10-30

**Soundness:** 3
**Presentation:** 3
**Contribution:** 2
**Rating:** 4
**Confidence:** 3

**Summary:**

The authors show that the Jacobians of robust networks act as implicit denoisers. In particular, they use a simple gradient ascent algorithm, Prior-Guided Drift Diffusion, to generate class-consistent samples from adversarially robust networks trained without generative objectives.

**Strengths:**

- The authors provide ample experimental evidence for their claims, both quantitative and qualitative.
- The work provides additional insights on robust classifiers

**Weaknesses:**

- The contribution is mild, as generation from adversarially robust classifiers was previously observed in Santurkar et al. (2019, https://arxiv.org/pdf/1906.09453). Differentiating their objective reveals a very similar algorithm to PGDD, as the input-derivative of scalar outputs of a network all strongly involve the jacobian.
- In-depth analysis is limited to Cifar-10. It would strengthen the paper if this was done for larger datasets like ImageNet. Perhaps the computational cost is too large for full-resolution, but for artificially downsampled images it should be possible.
- Image plausibility is much weaker for standard networks, which limits the contribution in a novel area i.e. beyond robust networks.
- Normalized eigenvalue statistics are not provided (e.g. Table 1) beyond lambda_1 / lambda_2. It would strengthen the analysis if such statistics were reported, such as soft-rank (normalized eigenvalue entropy) and normalized-eigenvalue plots comparing standard and robust networks.

**Questions:**

- Are there greater connections between denoising diffusion models and the denoising capabilities of adversarially robust networks? Any further insight connecting with modern generative models would strengthen the work.
- The authors might consider interesting the work of Rodriguez-Munoz et al. (2024, https://arxiv.org/pdf/2409.20139), which showed a causal relationship of spectral properties -> robustness, as mentioned in the limitations section of the paper. In particular, Section 5.3 shows that aligning jacobians with natural image edges achieves robustness up to 60% of adversarial training.

---

> ### Author Response · Authors · 2025-11-23
> **Part 1**
>
> Thank you for this thorough and balanced review. We greatly appreciate your recognition that our work provides **"ample experimental evidence for [our] claims, both quantitative and qualitative"**. We are also grateful for your positive assessment of our work's soundness and presentation quality.
>
> **We have substantially strengthened both the theoretical and empirical components of the paper in response to your feedback.** Specifically, we have added formal theorems establishing theoretical rigor, expanded our in-depth analysis beyond CIFAR-10 to include systematic spectral analysis across MNIST, CIFAR-10, and ImageNet, included the requested normalized eigenvalue statistics (soft-rank and eigenvalue plots), and provided detailed connections to modern generative models including denoising diffusion models. We have also clarified our contribution relative to prior work, emphasizing our novel theoretical framework and mechanistic understanding.
>
> Below, we address each of your specific concerns systematically:
>
>
>
>
> #### 1. **Similarities to Santurkar et al. (2019)**
> > The contribution is mild, as generation from adversarially robust classifiers was previously observed in Santurkar et al. (2019, [https://arxiv.org/pdf/1906.09453](https://arxiv.org/pdf/1906.09453)). Differentiating their objective reveals a very similar algorithm to PGDD, as the input-derivative of scalar outputs of a network all strongly involve the jacobian.
>
> Thank you for raising this important point. As we state in the introduction, our work is directly inspired by the seminal observations of Santurkar et al. (2019) and related follow-up work, which demonstrated that adversarially robust classifiers exhibit surprising generative behavior. We fully agree that their gradient-based procedure also involves the Jacobian, and our results are consistent with that insight.
> However, our contribution is complementary in two key ways:
>
> **(1) Prior work documents that robust models generate, whereas our goal is to explain why.**
>
> Earlier methods showed that robust classifiers exhibit generative behavior but did not provide a mechanistic account of what property of robust training makes this possible. Our theoretical analysis identifies the Gram operator J^\top J as the key object and shows how adversarial training induces the low-rank, spectrally concentrated structure that enables generation. PGDD is introduced to make this mechanism explicit: it reveals that the denoising behavior of J^\top J is the driver behind the generative capability.
> **(2) PGDD performs unconditional generation, unlike prior conditional methods.**
>
> The generation procedures in Santurkar et al. and related papers require a **target**—either a class label or a processed reference image—so they perform _targeted_ or _conditional_ synthesis.
>
> In contrast, PGDD relies **only on the input image itself** and generates patterns that arise solely from the structure of J^\top J, without specifying any target class or desired output. This makes PGDD a fundamentally different and **unconditional** generative mechanism, directly tied to the model’s internal spectral structure rather than to externally supplied targets.
> **(3) Our framework unifies and explains why different gradient-based visualizations work.**
>
> The fact that the derivative of many visualization objectives includes the Jacobian is consistent with our analysis: we show that the _spectral properties_ of the Jacobian—specifically the concentration of the Gram operator—are what enable robust models to yield coherent generative patterns across diverse gradient-based procedures. PGDD is therefore not intended as a replacement for these methods, but as a principled mechanism that explains why they succeed in the robust setting.
>
> We have clarified this connection in the revised manuscript (Introduction) and are happy to expand on it further if helpful.

---

> > ### Author Response · Authors · 2025-11-23
> > **Part 2**
> >
> > ### 2. **Spectral analysis at the ImageNet level**
> > > In-depth analysis is limited to Cifar-10. It would strengthen the paper if this was done for larger datasets like ImageNet. Perhaps the computational cost is too large for full-resolution, but for artificially downsampled images it should be possible.**
> >
> > Thank you for this helpful suggestion. In the revised manuscript, we have substantially expanded our spectral analysis to include **full-resolution ImageNet models** (ResNet-50, $\ell_2$-robust with $\epsilon\in{0.00, 3.00, 5.00}$). While computing the complete spectrum for ImageNet is indeed computationally prohibitive, we implemented efficient approximations—Hutchinson trace estimation, subspace iteration for top-$k$ eigenvalues, energy-ratio statistics, and soft-rank—to capture the same spectral trends observed at the CIFAR-10 scale.
> >
> > These additions include:
> >
> > -   **Visual residuals** ($J^\top J\varepsilon$) at 224×224 resolution, showing that robust ImageNet models continue to extract coherent, class-structured patterns from noise (Fig. S3 in the Supplement).
> >
> > -   **Top-$k$ normalized eigenvalue decay curves** for ImageNet (Fig. S4), demonstrating the same strong spectral concentration seen at smaller scales.
> >
> > -   **Soft-rank and entropy-based effective dimensionality**, confirming that adversarial training collapses $J^\top J$ to a low-dimensional subspace even for high-resolution models.
> >
> > -   **Energy-ratio distributions** and **PGDD PSNR improvements** on ImageNet, quantitatively verifying that robust ImageNet classifiers exhibit strong denoising structure whereas standard ImageNet models do not (Table 4 in supplementary).
> >
> > We thank the reviewer for this suggestion; adding these analyses has made the empirical support for our theoretical claims substantially stronger.
> >
> > ### 3. **Weak generation in standard networks**
> > > Image plausibility is much weaker for standard networks, which limits the contribution in a novel area i.e. beyond robust networks.
> >
> > We agree that the images generated from standard (non-robust) networks are much weaker than those produced by robust models. Importantly, our goal was never to demonstrate high-fidelity generation from standard classifiers. Instead, the purpose of sPGDD was to **empirically validate a theoretical prediction**:
> >
> > > **If standard networks possess even a very weak version of the implicit denoising structure found in robust models, then applying suitable smoothing should amplify this weak signal into faint—but nontrivial—generative behavior.**
> >
> > This prediction is borne out in our experiments.
> >
> > ### **1. Weak generation in standard models is an intended prediction, not a limitation**
> >
> > Our theoretical framework predicts that:
> > -   robust training produces a **strong, low-rank, class-aligned** denoising operator J^\top J,
> > -   while standard networks contain only a **very weak and noisy** implicit structure **due to implicit regularization (e.g., Barrett & Dherin 2020)** (Drawing inspiration from work showing that gradient descent implicitly regularizes parameters (Barrett & Dherin 2020), we hypothesize that similar regularization effects may weakly structure input gradients in standard networks.).
> > Thus, the difference in generation quality is **expected** and precisely matches our theory.
> >
> > ### **2. The contribution is not “generating from standard networks,” but showing the existence of generative structure**
> >
> > Although the samples from standard models are low-fidelity, they nonetheless demonstrate something genuinely new:
> >
> > -   Prior work on generative abilities of classifiers has focused exclusively on _robust_ models.
> >
> > -   > To our knowledge, **no existing method—SmoothGrad, saliency visualization, or classifier-based generative modeling—has produced generative outputs from standard networks**.
> >
> > -   Even weak generation therefore constitutes **new empirical evidence** supporting our theoretical link between the Gram operator and generative behavior.
> >
> > ### **3. sPGDD is introduced only as a probe into this structure, not as a generative method**
> >
> > We emphasize that:
> >
> > -   PGDD is the principled generative mechanism predicted by our theory.
> >
> > -   sPGDD is an _auxiliary tool_ designed solely to reveal faint structure in standard models.
> >
> > -   It is not meant to be a practical generative algorithm.
> >
> > Thus, the weaker outputs are not a failure of sPGDD—they are a **validation** of our theoretical expectation.
> > In summary,  the weaker generation observed in standard networks is not meant to compete with the outputs of robust models. Rather, it provides the first empirical evidence that even standard classifiers contain a very weak implicit denoising structure—which sPGDD can amplify—supporting our theoretical connection between the Gram operator and generation. No prior work has demonstrated such behavior (a method as simple as smoothgrad can generate), and this constitutes a novel contribution beyond existing results on robust models.

---

> > > ### Author Response · Authors · 2025-11-23
> > > **Part 3**
> > >
> > > ### 4. Further spectral analyses
> > > >Normalized eigenvalue statistics are not provided (e.g. Table 1) beyond lambda_1 / lambda_2. It would strengthen the analysis if such statistics were reported, such as soft-rank (normalized eigenvalue entropy) and normalized-eigenvalue plots comparing standard and robust networks.
> > >
> > > Thank you for the suggestion. We have added all the requested spectral analyses to strengthen the claims:
> > > -   **Normalized eigenvalue decay plots** (λᵢ/λ₁) are now included for MNIST, CIFAR-10, and ImageNet (Figures S4–S6).
> > >
> > > -   **Soft-rank (entropy-based effective rank)** is now reported for all models in the updated spectral tables, including ImageNet (Table 4).
> > >
> > > These additions show a consistent pattern: robust models exhibit much steeper normalized eigenvalue decay and  lower soft-rank, confirming strong spectral concentration relative to standard models.
> > > >
> > > ### Questions
> > > >** Q1**. Are there greater connections between denoising diffusion models and the denoising capabilities of adversarially robust networks? Any further insight connecting with modern generative models would strengthen the work.
> > >
> > >
> > > Thank you for this insightful suggestion. Our revised draft now expands the discussion of how the denoising behavior of adversarially robust networks relates to modern generative modeling.
> > >
> > >
> > > First, we draw an explicit connection to recent work on **_compositional generative modeling_** (e.g., Du & Kaelbling, 2024). These approaches argue that a single monolithic generator is insufficient, and advocate instead for modular energy components that can be flexibly composed. A robust classifier naturally instantiates this structure: each class logit defines an implicit energy function, and PGDD allows sampling from any individual class or linear combinations of classes. Thus, a single adversarially trained model contains a rich family of compositional generative pathways, aligning closely with the goals of these recent methods.
> > >
> > > Second, recent analytic work (e.g., Kamb & Ganguli, 2024) shows that **diffusion models avoid degenerate memorization only when equipped with appropriate inductive biases** (e.g., locality, equivariance) that constrain the score function. In our setting, the class-conditioning inherent to a robust classifier plays a similar regularizing role: the classification objective forces the denoising dynamics to remain tied to a semantic class manifold rather than collapsing into memorization of training examples. This mechanism closely parallels how structured generative behavior emerges in diffusion models.
> > >
> > > We will incorporate these connections directly into the main text if the reviewer believes they would strengthen either the conceptual motivation or the broader implications of our work. Thanks again.
> > >
> > >
> > > >**Q2**. The authors might consider interesting the work of Rodriguez-Munoz et al. (2024, [https://arxiv.org/pdf/2409.20139](https://arxiv.org/pdf/2409.20139)), which showed a causal relationship of spectral properties -> robustness, as mentioned in the limitations section of the paper. In particular, Section 5.3 shows that aligning jacobians with natural image edges achieves robustness up to 60% of adversarial training.
> > >
> > > Thank you for pointing us to this relevant work. We have incorporated a citation to Rodríguez-Muñoz et al. (2024) in the revised manuscript in the section discussing Jacobian-based regularization and spectral mechanisms of robustness. This work aligns well with our related work section, as it provides causal evidence that explicitly shaping Jacobian spectral structure can improve adversarial robustness

---

### Official Review · Reviewer_bcZP · 2025-10-31

**Soundness:** 2
**Presentation:** 2
**Contribution:** 2
**Rating:** 4
**Confidence:** 2

**Summary:**

This paper proposes a theoretical framework to explain the emergent generative capabilities of adversarially robust classifiers. The central hypothesis is that adversarial training induces a low-rank structure in the network's input-output Jacobian, $J$. Consequently, the Gram operator, $J^TJ$, functions as an implicit denoiser, selectively preserving signal along a low-dimensional "discriminative subspace" while suppressing noise in orthogonal directions.

To leverage this insight, the authors introduce Prior-Guided Drift Diffusion (PGDD), a novel inference-time algorithm that generates images without requiring any generative training or architectural changes. PGDD works by iteratively updating an input to move its internal representation away from that of a noisy version of itself. The authors show that the gradient of this objective elegantly approximates a denoising step involving $-J^T J$. They extend this method to standard (non-robust) networks with smooth PGDD (sPGDD), which uses gradient averaging to stabilize the process.

**Strengths:**

- The paper's core contribution is providing a simple and principled explanation for a well-documented but poorly understood phenomenon. The identification of $J^T J$ as an implicit denoiser connects the spectral properties of robust networks (low-rank Jacobians) with the mechanisms of modern generative models (denoising).
- The paper is well-written and easy to follow. The introduction clearly motivates the problem and outlines the contributions.

**Weaknesses:**

1. Limited Scope of Full Spectral Analysis: While the spectral analysis on MNIST is excellent, it is limited to a small-scale problem. The paper's central theoretical claim about the $J^T J$ operator is not directly verified for the large-scale ImageNet models due to the computational intractability of computing the Jacobian. The success of PGDD on these models provides strong indirect evidence, but the paper relies on the assumption that the same low-rank spectral properties hold.

2. Lack of Quantitative Generative Metrics: The paper relies entirely on qualitative visual inspection for the generated ImageNet samples. While the images are coherent, their quality is visibly lower than state-of-the-art generative models. Including standard quantitative metrics like FID or Inception Score would be beneficial. Even if not state-of-the-art, these scores would provide a concrete way to compare generation quality across different models architectures, and between PGDD and sPGDD.

**Questions:**

1. Could you comment on the computational cost of PGDD/sPGDD for generating a single image compared to, for example, a forward pass in a comparably sized DDPM or GAN?

2. How were the hyper-parameters in Tables 3 and 4 selected? Was it manual tuning, or is there a more systematic procedure? For example, how does the generation trajectory change with different values of the Drift Noise Ratio or Step Size? Do you observe any failure modes (e.g., divergence, mode collapse) with improper settings?

3. While acknowledging the quality is not SOTA, could you provide FID/IS scores for the generated ImageNet samples? This would help quantitatively benchmark the effect of different robustness levels (ϵ), architectures, and the difference between PGDD on robust models vs. sPGDD on standard models.

4. Can you further elaborate on the theoretical relationship between the PGDD update direction $\nabla_x$LPGDD ≈ $-J^T J \epsilon$ and the score function $\nabla_x \log p(x)$? Is it possible to view PGDD as a form of score matching or Langevin dynamics on an implicit energy function defined by the classifier?

---

> ### Author Response · Authors · 2025-11-23
> **Part 1**
>
> Thank you for this thorough and constructive review. We greatly appreciate your recognition of our work's core contribution—**providing a simple and principled explanation for the well-documented but poorly understood generative capabilities of robust classifiers through the identification of J^T J as an implicit denoiser**. We are also grateful for your positive assessment of the paper's clarity and motivation.
> Below, we address each of your specific concerns and questions systematically:
> ### 1. Limited Scope of Full Spectral Analyses
> > While the spectral analysis on MNIST is excellent, it is limited to a small-scale problem. The paper's central theoretical claim about the operator is not directly verified for the large-scale ImageNet models due to the computational intractability of computing the Jacobian. The success of PGDD on these models provides strong indirect evidence, but the paper relies on the assumption that the same low-rank spectral properties hold.
> **Systematic spectral verification across scales (MNIST, CIFAR10, and ImageNet):** We have now used efficient spectral probing methods that **directly verify our central theoretical claims about the J^T J operator across all three scales**. **We now provide systematic spectral analysis across robustness levels ε ∈ {0.00, 0.25, 0.50, 0.75, 1.00} for MNIST, ε ∈ {0.00, 0.25, 0.50, 1.00} for CIFAR-10, and ε ∈ {0.00, 3.00, 5.00} for ImageNet** using Hutchinson trace estimation and implicit Jacobian-vector products that avoid explicit matrix construction.
> ** Our expanded analysis demonstrates the **same fundamental spectral concentration patterns hold across all scales**:
> - Trace tr(J^T J) shows systematic reduction with increased robustness (127.1 → 23.8 for MNIST, similar patterns for CIFAR-10/ImageNet)
> - Energy ratio ||J^T J ε||/||ε|| decreases consistently
> - R² correlations improve dramatically, confirming structured spectral relationships
>  The **systematic spectral concentration phenomenon holds from simple MNIST digits to ImageNet natural images**, demonstrating this is a fundamental property of adversarial training rather than a small-scale artifact. Visual residual analysis across all three datasets shows the J^T J operator transitions systematically from noise-preserving to pattern-extracting behavior.
>
> ### 2. Lack of Quantitative Generative Metrics
> > The paper relies entirely on qualitative visual inspection for the generated ImageNet samples. While the images are coherent, their quality is visibly lower than state-of-the-art generative models. Including standard quantitative metrics like FID or Inception Score would be beneficial. Even if not state-of-the-art, these scores would provide a concrete way to compare generation quality across different models architectures, and between PGDD and sPGDD.
> >
> >Q3. While acknowledging the quality is not SOTA, could you provide FID/IS scores for the generated ImageNet samples? This would help quantitatively benchmark the effect of different robustness levels (ϵ), architectures, and the difference between PGDD on robust models vs. sPGDD on standard models.
>
> We acknowledge this limitation and agree that quantitative metrics would strengthen our analysis. We emphasize that generation quality in our work will never approach models explicitly trained for generation. **Our primary contribution in developing PGDD as an empirical tool was to validate our main theoretical goal: **understanding WHY robust classifiers exhibit (though modest) generative properties**. That said, **it is nonetheless informative to systematically compare across different classifiers and robustness levels** to understand how generated patterns relate to real images. Such comparisons can validate whether stronger spectral concentration correlates with better generation quality, supporting our theoretical framework.
>
> **MNIST FID analysis:** To address this need, **we require substantial numbers of generated samples for reliable metric computation**. We conducted comprehensive PGDD generation on MNIST models and developed an appropriate FID score evaluation suitable for the MNIST domain. [**section in supplementary: FID Score for evaluation of generated patterns**]
> The FID analysis revealed **non-trivial relationships between robustness levels and generation quality—rather than a simple "more robust = better generation" pattern, we found that the choice of reference model significantly affects scores, with evidence of asymmetric feature space relationships between different robustness levels.** For example, ε=1.00 generated images achieved better scores when evaluated by clean and ε=0.25 reference models, but ε=0.25 generated images performed best when evaluated by the ε=1.00 reference model, suggesting that evaluating classifier-generated patterns is not as straightforward as evaluating patterns from dedicated generative models due to interactions between embedding directions and the generated patterns.

---

> > ### Author Response · Authors · 2025-11-23
> > **Part 2**
> >
> > **Computational constraints for larger scales:** For CIFAR-10 and ImageNet, the computational cost of generating sufficient samples (thousands needed for reliable FID computation) with our iterative PGDD process remains a practical limitation (sPGDD reuires x100 more). However, the MNIST analysis demonstrates the feasibility of quantitative evaluation and provides a foundation for systematic comparison across robustness levels and architectures.
> >
> > ### Questions
> > >Q1. Could you comment on the computational cost of PGDD/sPGDD for generating a single image compared to, for example, a forward pass in a comparably sized DDPM or GAN?
> > >
> > **PGDD typically requires ~500-1000 iterations per image for ImageNet models** (each with one forward + backward pass through the classifier), making it **computationally comparable to DDPMs** (~1000 U-Net forward passes) but **significantly slower than GANs** (single forward pass).
> >
> > Computational trade-offs:
> > - vs. DDPMs: Similar inference cost but zero generative training required albeit generated patterns quality are not matched
> > - vs. GANs: 1000× slower inference but leverages existing robust classifiers
> > - sPGDD: 100× more expensive than PGDD due to gradient smoothing (~100 samples/iteration)
> >
> >
> > >Q2. How were the hyper-parameters in Tables 3 and 4 selected? Was it manual tuning, or is there a more systematic procedure? For example, how does the generation trajectory change with different values of the Drift Noise Ratio or Step Size? Do you observe any failure modes (e.g., divergence, mode collapse) with improper settings?
> >
> > Parameter selection was based on simple manual exploration rather than exhaustive optimization, given our focus on demonstrating the underlying mechanism rather than achieving optimal generation quality.
> >
> > **Systematic parameter sensitivity analysis:**
> > - **Drift Noise Ratio**: Effective range ~0.005-0.02. Values below 0.005 provide insufficient stochasticity for exploration, while values above 0.05 cause noise to dominate, degrading semantic structure.
> > - **Step Size**: Optimal range 0.1-1.0, balanced with iteration count. Smaller values require more iterations for convergence; larger values (>1.0) produce high-contrast, exaggerated patterns that sacrifice semantic fidelity.
> > - **Iteration Count**: Typically 50-1000 depending on step size and convergence criteria.
> >
> > **Observed failure modes:**
> > - **Divergence**: High step sizes (>2.0) or excessive drift noise (>0.1) cause trajectories to diverge from semantic attractors
> > - **Stagnation**: Very small step sizes (<0.05) result in minimal pattern evolution
> > - **Noise dominance**: High drift noise ratios overwhelm the implicit denoising structure
> >
> >
> > **Generalization across architectures**: The same parameter ranges remain effective across different network architectures (ResNet, VGG, ViT), suggesting robustness of the underlying mechanism.
> >
> > >**Q4**.Can you further elaborate on the theoretical relationship between the PGDD update direction LPGDD ≈ and the score function ? Is it possible to view PGDD as a form of score matching or Langevin dynamics on an implicit energy function defined by the classifier?
> >
> > This is indeed an elegant theoretical framing that captures the core intuition behind our work.
> >
> > Conceptual alignment with score matching: PGDD does exhibit several key properties analogous to score-based sampling:
> > - **Iterative refinement**: Both PGDD and Langevin dynamics perform iterative updates guided by learned statistical regularities
> > - **Noise-to-structure progression**: Both frameworks transform random noise into structured patterns through gradient-based navigation
> > - **Implicit energy landscapes**: The PGDD objective effectively defines an energy function where robust classifiers create low-energy regions corresponding to class-consistent patterns
> >
> > **Theoretical challenges for formal equivalence:** However, as you astutely observe, **the structural dependence of J on both input x and class labels creates fundamental complications** for establishing direct equivalence:
> > - **Class-conditional structure**: The implicit energy landscape is inherently multi-modal (one mode per class), unlike typical score matching settings
> > - **Non-stationary dynamics**: The update directions change as x evolves during PGDD iteration
> >
> > Formalizing this connection remains an important direction, potentially through frameworks that handle input-dependent, multi-modal score functions or by developing theoretical tools for analyzing class-conditional implicit energy landscapes.
> >
> >
> > Are there any other aspects of the work that would benefit from further clarification or analysis?
> > Thank you again for your valuable and constructive feedback.

---

> ### Comment · Reviewer_bcZP · 2025-11-27
>
> Thank you for the detailed replies; most of my concerns have now been addressed. I have a few remaining points regarding the paper's presentation:
>
> 1. How do you derive equation 7 from equation 6?
> 2. The notation for the model input is inconsistent. For example, x is used in some places (Eq. 9, 10), while **x** is used when introducing PGDD. Please unify this notation.
> 3. The presentation of PGDD and sPGDD lacks a smooth transition. I suggest you explicitly state the update rule for **x** at different steps when introducing PGDD, similar to the format of Equation 10.
> 4. The function r(·) is defined but not used in the experiments. Please specify what was used for r(·) during experiments, for instance in the ImageNet robust model experiments.
> 5. To improve the readability of Section 4.2, I recommend starting each paragraph with its main conclusion presented as a bolded sentence.

---

> > ### Author Response · Authors · 2025-11-27
> >
> > We sincerely thank the reviewer for their thorough engagement with our work and for providing these specific, constructive suggestions. We addressed all five points in our revision:
> >
> > >1.  How do you derive equation 7 from equation 6?
> >
> > You correctly identified that this step needed clarification. The transition uses a first-order Taylor approximation that we failed to make explicit: r(x+ε) ≈ r(x) + J_r(x)ε.
> >
> > Since sg[·] only prevents gradient flow during backpropagation (not affecting forward values), substituting this approximation into equation (6) gives:
> >
> > ∇_x L_PGDD(x;ε) ≈ 2J_r(x)^T(r(x) - (r(x) + J_r(x)ε)) = -2J_r(x)^T J_r(x)ε
> >
> > We revised the manuscript to add "Using the first-order Taylor approximation r(x+ε) ≈ r(x) + J_r(x)ε:" between equations (6) and (7).
> >
> > >2.  The notation for the model input is inconsistent. For example, x is used in some places (Eq. 9, 10), while  **x**  is used when introducing PGDD. Please unify this notation.
> >
> > Thank you for pointing out this notation inconsistency. We have unified the notations in equations (5)-(8).
> >
> > >3. The presentation of PGDD and sPGDD lacks a smooth transition. I suggest you explicitly state the update rule for  **x**  at different steps when introducing PGDD, similar to the format of Equation 10.
> >
> > Thanks. We have made the PGDD update rule more explicit by clearly stating the iterative nature of the algorithm in equation (8), paralleling the presentation format used for sPGDD in equation (10). This improves the transition between PGDD and sPGDD sections.
> >
> > >4.  The function r(·) is defined but not used in the experiments. Please specify what was used for r(·) during experiments, for instance in the ImageNet robust model experiments.
> >
> > Thank you for pointing to this lack of clarity. We have added a clarification at the beginning of Section 4 stating that r(·) represents the logits (pre-softmax) layer throughout all experiments unless otherwise stated.
> >
> > >5.  To improve the readability of Section 4.2, I recommend starting each paragraph with its main conclusion presented as a bolded sentence.
> > >
> > That's a great suggestion, and we agree that it improves the readability. We now updated Section 4.2 with a short main conclusion sentence at the beginning of each paragraph.
> >
> > Thank you again for really helpful suggestions, the manuscript is now improved in readability and consistency. We are, of course, happy to clarify any remaining issues or incorporate additional suggestions as needed.

---

> ### Comment · Reviewer_bcZP · 2025-11-28
>
> Thank you for your responses. It appears the notation for the input has not yet been fully corrected. For example, x is defined or used in lines 177-178, 305-306, and 403-406, while a bold version **x** is used elsewhere.
>
> Could you please clarify if this distinction is intentional? If not, I would recommend unifying the notation for consistency.

---

> > ### Author Response · Authors · 2025-11-28
> >
> > Thank you for pointing this out. The distinction was indeed not intentional. We have now unified all occurrences of the input variable to use the boldface notation $\mathbf{x}$ throughout the paper, including the instances you noted (lines 177–178, 305–306, 403–406) as well as several additional locations we identified during a full pass of the manuscript.

---

### Official Review · Reviewer_WvWt · 2025-10-31

**Soundness:** 3
**Presentation:** 3
**Contribution:** 3
**Rating:** 6
**Confidence:** 3

**Summary:**

This study provides a theoretical perspective on why adversarial training elicits generative properties in robust image classifiers. It is demonstrated that the Jacobian Gram operator functions as an implicit denoiser, and an algorithm called Prior-Guided Drift Diffusion (PGDD) is developed to generate images. Furthermore, a sPGDD variant is proposed for standard (non-robust) classifiers by computing gradients over multiple independent perturbations of the inputs (akin to SmoothGrad). The study completes with empirical results aimed to demonstrate the implicit denoiser theory (on MNIST classifiers) and generations on ImageNet and MNIST classifiers.

**Strengths:**

- The works provides a novel (to the best of my knowledge) link between adversarial training and generative properties by formulating the implicit denoiser theory
- The authors propose PGDD to generate images by avoiding to compute $J^{\top}J$ explicitely.
- PGDD is further extended to sPGDD allowing generation via non-robust classifiers.
- The theory is (partially) backed by empirical evidence
- The limitation section is through and upfront

**Weaknesses:**

- The empirical evidence through the spectral analysis is very thin and only builds on one pair of networks (MNIST, $L_2$ and $\epsilon=1.5$, PGD-AT). If computing the Jacobian on ImageNet is too expensive, the paper could establish more evidence by recording the metrics as a function of $\epsilon$ where there should be a correlation.
- In similar fashion, the "visual residuals" analysis shown in Fig. 1 is thin by building on just one pair of images. Maybe this is a sign of my pareidolia, but I do see a rather clear "7" and opposed to the rather noisier sample of the robust network. It would be useful to show more examples (e.g., in the appendix).
- It is unclear how well these findings generalize to other forms of AT (e.g. as simple as FGSM or more complex), norms, network architectures (CNN vs. ViT) and datasets.
- *Is this really generation?* The T-SNE plots are used as argument for generation rather than membership inference (memorization). I am not convinced that this is sufficient. The generated images lie different manifolds, but style-wise they are quite different, which may be the actual underlying cause. It would be great to show a few examples of generated images to the nearest training samples in a) image space and b) latent space.
- While I appreciate how upfront the limitations section is, I am still concerned about some of the limitations. E.g., L403 states “multiple runs frequently arrive at the same predicted class”. This may be conceived by a failure of PGDD to fully reconstruct all classes. Is there any evidence that PGDD can "reach" all classes?

**Questions:**

See above and additionally:
- It seems like there are some "attractor" samples which PGDD collapses to.  What is special about these samples or their classes? Is there a higher accuracy for them or any other correlation that the authors could establish?
- Is there a difference between and $L_\inf$ training?

---

> ### Author Response · Authors · 2025-11-23
> **Part 1**
>
> We thank the reviewer for their thoughtful and constructive feedback, including the important question regarding the nature of the generated patterns. We appreciate the recognition of the **novelty** of this work.
> We have revised the manuscript extensively in response to the comments, adding deeper spectral analyses, broader empirical validation across datasets, and clarifying the theoretical claims.
> Below we address each point in detail.
>
>  ### 1. More empirical evidence through the spectral analysis
> > The empirical evidence through the spectral analysis is very thin and only builds on one pair of networks (MNIST, and , PGD-AT). If computing the Jacobian on ImageNet is too expensive, the paper could establish more evidence by recording the metrics as a function of where there should be a correlation
>  > In similar fashion, the "visual residuals" analysis shown in Fig. 1 is thin by building on just one pair of images. Maybe this is a sign of my pareidolia, but I do see a rather clear "7" and opposed to the rather noisier sample of the robust network. It would be useful to show more examples (e.g., in the appendix)**
>
>
> ### Added sweep over epsilon robustness values for spectral analysis and expanded visual residuals across three datasets
>
> We have comprehensively addressed both empirical concerns by expanding our analysis across multiple datasets and robustness levels.
>
> #### Systematic spectral analysis across robustness levels
>
> **We provide systematic spectral analysis across robustness levels ε ∈ {0.00, 0.25, 0.50, 0.75, 1.00} for MNIST, ε ∈ {0.00, 0.25, 0.50, 1.00} for CIFAR-10, and ε ∈ {0.00, 3.00, 5.00} for ImageNet (Section on spectral properties).** The comprehensive analysis reveals **systematic correlations** between adversarial training strength and spectral properties exactly as the reviewer requested:
>
> 1. **Quantitative correlations**: The spectral properties table demonstrates monotonic relationships between ε and key spectral metrics—trace tr(J^T J) decreases from 127.1 to 23.8 (5× reduction), energy ratio ||J^T J ε||/||ε|| decreases from 1.18 to 0.25 (5× reduction), and R² correlation improves from 0.247 to 0.999, showing increasingly structured spectral relationships.
>
> 2. **Cross-dataset consistency**: The same spectral concentration phenomenon holds from simple MNIST digits to complex ImageNet natural images, demonstrating that this is a fundamental property of adversarial training rather than a dataset-specific artifact.
>
> #### Expanded visual residuals analysis
>
> **Added visual residuals for more models across three datasets:** We have now included visual residual analyses for multiple MNIST models with varying robustness levels, as well as CIFAR-10 models tested on 5 different input noise patterns, and ImageNet models. **We agree with the reviewer that residuals in MNIST-trained standard networks are not completely random and can show digit-like patterns (this is not the case in the CIFAR-10 or ImageNet models)**. We believe this arises from the fact that standard and robust training on MNIST achieve such similar performance levels that differences are harder to differentiate. **Thus, including CIFAR-10 and ImageNet analyses is even more essential, and we thank the reviewer for suggesting them.**
>
> The expanded visual evidence across scales provides validation that the J^T J operator transitions systematically from noise-preserving (ε=0.00) to pattern-extracting (higher ε values) across all three datasets and architectures. This substantially strengthens our empirical foundation beyond the original single-model comparison, directly addressing both reviewer concerns with the requested correlation analysis and multiple visual examples.
>
> **Paper updated with revised Figure 1 now showing empirical validation across MNIST, CIFAR-10, and ImageNet datasets, and comprehensive supplementary section (Section on spectral properties) with detailed spectral analysis across robustness levels for all three datasets.**

---

> > ### Author Response · Authors · 2025-11-23
> > **Part 2**
> >
> > ### 3. Other form of AT
> >  > It is unclear how well these findings generalize to other forms of AT (e.g. as simple as FGSM or more complex), norms, network architectures (CNN vs. ViT) and datasets.
> >
> > Thank you for suggesting these analyses. We have systematically expanded our evaluation to address generalization concerns across multiple dimensions: **Adversarial training methods and norms:** We have now added PGDD generation results for models trained with both L2 and L∞ adversarial training. Specifically, we include ResNet-50 models trained with L∞ PGD (ε=4 and ε=8) in addition to our original L2 PGD models, demonstrating that the implicit denoising structure emerges regardless of the specific adversarial norm used during training. **Network architectures:** We have expanded our analysis across diverse architectures including Vision Transformers (ViT-S), convolutional networks of varying depths (VGG16-bn, ResNet-18), and wider architectures (ResNeXt-50-32x4d). This architectural diversity demonstrates that spectral concentration and implicit denoising capabilities are not specific to particular network designs but represent a fundamental property of robust training. **Cross-architecture consistency:** The generation results show that PGDD successfully produces coherent semantic content across all tested architectures, with the quality and convergence patterns remaining consistent. This suggests that the underlying mathematical principle—spectral concentration in the Jacobian structure—is architecture-agnostic. While we focused on PGD-based adversarial training in this work, the mathematical framework predicts that any training method that induces spectral concentration should exhibit similar implicit denoising properties. We acknowledge that comprehensive evaluation across all adversarial training variants (FGSM, C&W, etc.) represents valuable future work that would further strengthen the generality claims.
> > > Is there a difference between L2 and Linf training?
> >
> > ### Differences between L2 and L∞ adversarial training
> >
> > We systematically compared PGDD generation capabilities across L2 (ε=3,5) and L∞ (ε=4,8) adversarial training:
> >
> > **Important methodological note:** We acknowledge that our L2 and L∞ models use different epsilon values—L2 trained with ε=3,5 while L∞ models (ε=4,8) were obtained from Madry's robustness package. **This makes direct quantitative comparison challenging**, as epsilon magnitude affects spectral concentration independent of norm choice. Nevertheless, our analysis provides valuable insights into the behavioral differences between these training paradigms.
> >
> > The spectral metrics show nuanced differences rather than clear superiority of one norm over another. L2 models achieve better performance in some metrics (e.g., L2 ε=3 vs L∞ ε=4: lower tr(J^T J) of 0.22 vs 0.49, and better PGDD ΔPSNR of 1.22dB vs 0.77dB), while L∞ models excel in others (e.g., L∞ ε=8 achieves the lowest trace value of 0.11 and highest PGDD ΔPSNR of 1.32dB).
> >
> >  Despite comparable or sometimes superior quantitative metrics, **L∞ trained models consistently produce less semantically coherent patterns** than L2 models. Visual assessment reveals that L2 generation exhibits clearer semantic structure and better pattern organization (Examples in Figure 3).
> >
> > **Theoretical insight on L2 vs L∞ performance:** An important theoretical consideration is that **successful denoising diffusion models universally employ Gaussian (L2) noise** in their forward processes. L2 adversarial training creates implicit denoising structure aligned with this same noise geometry, while L∞ training learns to handle max-norm perturbations that create a geometric mismatch with effective generative processes. This explains why L2 models produce more semantically coherent patterns despite mixed quantitative metrics—they learn denoising operations compatible with the noise structures that enable generation.
> > **Additionally, L∞ adversarial training typically provides (weaker) L2 robustness as well** (since defending against worst-case perturbations often generalizes to other norms), which explains why L∞ models still demonstrate some denoising capability and generation albeit with reduced semantic coherence.

---

> > > ### Author Response · Authors · 2025-11-23
> > > **Part 3**
> > >
> > > ### 4. Generation?
> > > > Is this really generation? The T-SNE plots are used as argument for generation rather than membership inference (memorization). I am not convinced that this is sufficient. The generated images lie different manifolds, but style-wise they are quite different, which may be the actual underlying cause. It would be great to show a few examples of generated images to the nearest training samples in a) image space and b) latent space.
> > >
> > > This is a very insightful question. We started our quest to look into generativity of adversarially robust neural network, taking their generativity power granted from numerous literature in the field  (examples: Santurkar et al. 2019 demonstrating image synthesis with robust classifiers; Engstrom et al. 2019 showing that robust models produce perceptually aligned gradients and improved correspondence between internal representations and human-perceivable features; Kaur et al. 2019 establishing that robust networks generate more interpretable visualizations;"). The generated images could be templates of categories, or even templates of differential between categories (e.g., deer vs horse in CIFAR10).
> > >
> > > **FID Analysis Results:** We have implemented domain-appropriate FID scores to quantitatively assess generation quality. For MNIST, we developed a modified FID score using TinyResNet feature extractors (rather than Inception, which is inappropriate for MNIST). For CIFAR-10, preliminary analysis of 500 generated samples yields FID scores around 290, which could be due to small relative number of high-confidence (as scored by the generating classifier).  The high FID scores are expected though, we expect modest generation quality from classifiers trained without generative objectives
> > > -Generating the thousands of samples typically required for reliable FID computation (with 1000+ PGDD iterations per sample) remains computationally challenging, but our preliminary results demonstrate the feasibility of quantitative evaluation
> > > **MNIST comprehensive analysis**: Our detailed MNIST FID analysis [section: FID SCORE FOR EVALUATION OF GENERATED PATTERNS] provides the complete quantitative framework.
> > >
> > > We emphasize that our goal is not achieving competitive generation quality, but rather **understanding why robust classifiers exhibit any generative capability at all**. The high absolute FID scores actually expected, as noted in the paper, however, we agree that such scores are essential to compare the generativity across classfiiers, that's what we currectly included across MNIST dataset (generating thousands of images systematically by sweeping input noise seed, PGDD parameters, and inference seed).
> > >
> > >
> > >
> > > ### Distinguishing generation from memorization or class templates: Understanding the nature of produced patterns
> > >
> > > This is a very insightful question that addresses a fundamental methodological concern in evaluating generative capabilities. We acknowledge that t-SNE separation alone is insufficient evidence for true generation versus sophisticated memorization.
> > >
> > > **Building on established observations:** The hypothesis of generative properties in robust classifiers originates from established literature, particularly work from the Madry group (Santurkar et al. 2019; Engstrom et al. 2019) and others (Kaur et al. 2019), which demonstrated that robust networks can produce "perceptually aligned" patterns. **We are not claiming true generation in the sense of novel content creation**. Rather, our primary research goal is to **understand the nature and mechanism behind these produced patterns**: Are they reproductions of training examples? Combinations of training examples? Templates that differentiate between learned classes?
> > >
> > > **Research focus on pattern analysis:** This study's main focus is to understand **why robust networks produce patterns that appear "perceptually aligned"** as frequently referenced in the literature, and to provide a theoretical framework explaining this phenomenon through spectral properties. The question of whether these patterns represent memorization, interpolation, or template generation is precisely what we seek to characterize through rigorous analysis.
> > >
> > >  We are conducting the requested nearest neighbor analysis in both: (a) pixel space using L2 distance, and (b) latent space using penultimate layer representations (MNIST experiment already done, please see below) . This will provide quantitative evidence for the nature of these patterns—whether they represent direct memorization, statistical combinations, or prototypical templates.

---

> > > > ### Author Response · Authors · 2025-11-23
> > > > **Part 4**
> > > >
> > > > ### 5. Generation of all classes
> > > > >While I appreciate how upfront the limitations section is, I am still concerned about some of the limitations. E.g., L403 states “multiple runs frequently arrive at the same predicted class”. This may be conceived by a failure of PGDD to fully reconstruct all classes. Is there any evidence that PGDD can "reach" all classes?
> > > > > It seems like there are some "attractor" samples which PGDD collapses to. What is special about these samples or their classes? Is there a higher accuracy for them or any other correlation that the authors could establish?
> > > >
> > > > This is an excellent question that gets to the heart of PGDD's generative capabilities and limitations. **We have conducted a comprehensive analysis to address these concerns (Section on generated digit distribution analysis).**
> > > >
> > > > **Evidence for class reachability:** Our systematic analysis across multiple MNIST models demonstrates that **PGDD can indeed reach all 10 digit classes**, though with significantly non-uniform frequency. While certain digits (notably 4 and 8) appear as attractors with 5-10 times higher generation frequency, **all classes remain accessible** through appropriate parameter selection and initialization.
> > > >
> > > > **Characterizing the attractor structure:** The supplementary analysis reveals several key insights about these attractors:
> > > > - **No correlation with classification accuracy**: Despite relatively uniform clean accuracy across digit classes (96-99%), generation frequencies vary dramatically, indicating that attractor strength is **not simply determined by discriminative performance** Digit 9 shows the lowest per-class accuracy (98.1%) and is also the weakest attractor, while digit 4 has similarly low accuracy (98.4%) yet emerges as one of the strongest attractors (>20% generation frequency)
> > > >
> > > > The non-uniform distribution suggests certain digit geometries create stronger attractors in the learned representation space—potentially reflecting intrinsic structural properties of how different classes are encoded. We still don't know the nature of these attractor like behavior.
> > > >
> > > > We recognize that the vast parameter space makes exhaustive reachability claims impossible. **As established in the related literature of generative capability of classifiers, we adopt the conventional notion of "generation" as producing meaningful patterns** rather than claiming complete distributional coverage. Our revised limitations section reflects these constraints while highlighting the systematic insights gained about the attractor structure of robust networks.
> > > > Thank you for the thorough review and constructive feedback. If any concerns remain unaddressed, we would greatly appreciate specific guidance on further modifications. Additionally, we are considering reorganizing the paper structure by moving the comprehensive spectral experiments to the main text and relocating the t-SNE and trajectory analyses to supplementary material- would this improve clarity and emphasize our core empirical contributions? We are committed to making any adjustments needed to strengthen the work.
> > > >
> > > > Thank you again for your valuable feedback.

---

> > > > > ### Comment · Reviewer_WvWt · 2025-11-28
> > > > >
> > > > > Thanks for the very thorough rebuttal: I'll provide a more detailed response soon.
> > > > > What is the status for the promised results for "4. Generation?". The visual residuals in S2 look very similar to a "cat" sample from the train set, so I suppose this will be indeed more of a membership inference or interpolation at best.

---

> > > > > > ### Author Response · Authors · 2025-11-28
> > > > > >
> > > > > > Thank you for the astute observation about the cat-like appearance in Figure S2. We appreciate your careful examination of our results and have added **Supplementary Section F** with a nearest-neighbor analysis to address this important concern.
> > > > > >
> > > > > > **Important distinction between J^TJ𝜖 and full PGDD:**
> > > > > >
> > > > > > -   The J^TJ𝜖 visualizations in Figure S2 are single-step applications meant to illustrate the operator's denoising properties on random noise
> > > > > > -   Full PGDD involves iterative application with stochastic diffusion noise at each step, producing diverse outputs rather than converging to a single memorized pattern
> > > > > > -   The cat-like structure you observed in J^TJ𝜖 is indeed striking, but represents the operator's immediate response to noise rather than the full generative process
> > > > > >
> > > > > > **Theoretical focus:** The primary contribution of this work is providing a theoretical framework explaining the generative capabilities observed in robust classifiers (building on Santurkar et al., 2019). The emergence of structured patterns from J^TJ validates our signal-noise decomposition theory (Theorem 2.1), demonstrating that robust training induces selective amplification of signal directions while suppressing noise.
> > > > > >
> > > > > > **Results from Section F (for the samples we tested):**
> > > > > >
> > > > > > -   Nearest-neighbor analysis on PGDD-generated CIFAR-10 samples (Figure S6) shows high pixel-space distances from closest training examples
> > > > > > -   Logit-space analysis similarly confirms large distances from training data
> > > > > > -   For all generated samples we evaluated, we did not observe direct reproduction or interpolation of individual training examples
> > > > > > -   This quantitative evidence supports that PGDD is not generating training samples
> > > > > >
> > > > > > **We acknowledge** that the space of possible generated patterns is vast, and some may bear similarity to training data. In future work, we plan to investigate whether frequently generated patterns (attractor patterns) correspond to the most discriminative features between classes, potentially using similar methods in Haim et al. (2022, arXiv:2206.07758). However, such comprehensive characterization of generated patterns is beyond the scope of this work, which focuses on providing the theoretical foundation for understanding why robust classifiers exhibit generative capabilities.
> > > > > >
> > > > > > Thank you again, and we welcome any further suggestions to improve the work.

---

### Author Response · Authors · 2025-11-25
**General Response to Reviewers by the authors**

We thank all reviewers for their thoughtful feedback. We are grateful that reviewers recognized key strengths: the novel theoretical framework connecting adversarial training to generative capabilities (Reviewer WvWt), the conceptually intuitive nature of PGDD (Reviewer BXDv), the empirical evidence supporting our claims (Reviewer y7U3), and our clear presentation of this poorly understood phenomenon (Reviewer bcZP).

In response to the feedback, we have undertaken revisions that address the core concerns. The most significant enhancement involves expanding our theoretical foundation with formal mathematical rigor. We introduced Theorem 1 ("Signal–Noise Decomposition in J^T J") with complete proofs, providing the mechanistic explanation for how spectral concentration induced by adversarial training leads to implicit denoising capabilities. This directly addresses Reviewer BXDv's concern about the lack of formal theorems.

We have expanded our empirical validation beyond the original MNIST-focused analysis that multiple reviewers found limiting. We now provide systematic spectral analysis across MNIST, CIFAR-10, and ImageNet with sweeps over robustness levels, demonstrating that spectral concentration patterns hold consistently from simple digits to complex natural images. The correlation between adversarial training strength and spectral properties that Reviewer WvWt requested is now established through quantitative metrics including trace reduction, energy ratio improvements, and R² correlations.

We addressed quantitative evaluation by implementing domain-appropriate FID-like scores for MNIST using TinyResNet feature extractors. While computational constraints limit extensive FID computation for larger datasets, our MNIST analysis demonstrates the feasibility of quantitative comparison and reveals relationships between robustness levels and generation quality that support our theoretical framework.

The notation and presentation concerns raised by Reviewer BXDv have been addressed through formal definitions of all key terms, enhanced algorithmic descriptions for both PGDD and sPGDD, and improved explanations connecting empirical measurements to theoretical predictions.

Regarding Reviewer BXDv's important conceptual question about the specificity of implicit denoising structure, we clarified the distinction between robust and standard networks. Robust classifiers exhibit strong, low-rank, class-aligned implicit denoising operators through adversarial training, while standard networks contain only weak, noisy versions of this structure. The success of sPGDD demonstrates that gradient smoothing can extract weak approximations of latent structure, producing the low-fidelity outputs exactly as predicted by our theory.

We have extended our analysis across different adversarial training methods (L2 and L∞ PGD), network architectures (ViT, ResNet, VGG, ResNeXt), and provided connections to modern generative modeling frameworks. We also expanded our analysis of generation versus memorization through nearest neighbor analysis and systematic class reachability studies.

The revised manuscript now provides the clarification on theoretical foundation, empirical validation across multiple scales, and algorithmic descriptions that reviewers requested. We remain committed to addressing any remaining concerns and welcome additional feedback that would further strengthen this work.

---

### Meta-Review · Area_Chair_oXHF · 2026-01-07

**Summary:**

This work provides preliminary evidence for a connection between adversarial robustness and generative modeling by demonstrating that robust classifiers appear to contain implicit denoising structure encoded in their Jacobian operators. The main concerns from reviewers are

1) Limited evidence for supporting the claim. The authors only use the MNIST dataset to justify their claim (in the first version), which is not enough.

2) Limited theoretical justification. Although the intuition is interesting, more theoretical justification can further enhance the paper.

3) Empirical limitation. As generative models, the generation quality is not SOTA.

4) The evaluation metric is missing (e.g., FID).

5) Computational overhead is heavy for the proposed method.

6) The paper is not easy to follow, and the presentation needs improvement.

**Reviewer Concerns:**

After the rebuttal, the authors provide substantial changes to the original submission. Based on the new results, concerns 1) to 4) mentioned above are addressed. However, there are still two remaining concerns and one concern from the AC.

Based on the rebuttal, it can be found that the sPGGD is 100x slower than DDPM, which is too high. Thus, it basically means that the proposed methods are not very practical in the generative models.

However, like the author claimed in the rebuttal, "Our primary contribution in developing PGDD as an empirical tool was to validate our main theoretical goal: understanding WHY robust classifiers exhibit (though modest) generative properties", the performance is not the key point of this paper. Thus, AC has a concern about the paper's presentation. If the authors really want to emphsize on the above contribution, the revision still needs major changes, meaning that the current revision is not ready for ICLR yet. Thus, I recommend rejection for this round.

**Reviewer Scores:**

The authors indeed addressed many reviewers' concerns. As such,

- Reviewer WvWt should maintain the positive score of 6.
- Reviewer bcZP should maintain the score of 4, due to the SOTA concerns and the position of this paper (like the above comments)
- Reviewer y7U3's concerns are addressed and should increase the score from 4 to 6.
- Reviewer BXDv's concerns are addressed and clearly improved the score from 4 to 6.

Thus, the final score should be 6,4,6,6. However, based on comments in the box of Reviewer Concerns, I recommend rejection of this paper, due to major changes required in the current revision.

---

### Decision · Program_Chairs · 2026-01-26

Reject